



**Spatial and temporal variation in long-term temperature and water vapor in the**
**mesopause Region**
Chaman Gul*[1,2,3] , Shichang Kang[3], Yuanjian Yang [1], Xinlei Ge[2,4],  Dong Guo*[2]
[1]School of Atmospheric Physics, Nanjing University of Information Science & Technology, Nanjing, Jiangsu,
10044 China
[2]Reading Academy, Nanjing University of Information Science & Technology, Nanjing, Jiangsu, 210044 China
[3]Key Laboratory of Cryospheric Science and Frozen Soil Engineering, Northwest Institute of Eco-Environment
and Resources, Chinese Academy of Sciences, Lanzhou 73000, China
[4]School of Environmental Science and Engineering, Nanjing University of Information Science and
Technology, Nanjing, 210044, China
*Correspondence to:*
Chaman Gul* (gulchamangul76@yahoo.com) and Guo Dong* (002344@nuist.edu.cn)
Abstract:
Mesopause is the zone of minimum temperature in Earth's atmosphere. Temperature
variation in this region is one of the important responsible factors for chemical and physical
changes including spatiotemporal variability in water vapor content. Twenty-two years of
monthly temperature and water vapor data were used from Sounding of the Atmosphere
using Broadband Emission Radiometry. Eight months per year (excluding transitional
months) were selected for temporal analysis. Spatially the region is classified into four parts
including Northern, and Southern Poles. Long-term variations in water vapor and temperature
in the selected domains of time and space as well as at equinoxes and solstices are presented.
A decreasing, and increasing trend in temperature and water vapor respectively was observed
during the study period. Yearly averaged temperature and water vapor content showed that
2002 was the hottest year (193K) and had minimum water vaper content (0.89ppmv) and
2018 was the coldest year (187K) and had maximum water vapor content (1.14ppmv). June
and July were the coldest months and January and December were hotter months throughout
the year over the North Pole and Equator. The vertical gradient of temperature and water
vapor (80 to 100km) changes with space and time however, has a strong negative relation in
all selected locations and seasons. Around the equinoxes, the monthly average distribution of
mesopause temperature was highest (191K), followed by winter solstice and then summer
solstice. The decreasing trend in temperature and an increasing trend in water vapor can be an
early warning indication for future climate change.
Keywords: Atmosphere, Mesopause; Temperature; Water vapor; spatial and temporal changes.





## 1. Introduction

Mesopause is one of the complex and intricate domain regions of Earth's atmosphere. It is the thermal transition area that plays an important role in the vertical coupling of the Earth's atmosphere. In the global mean temperature, mesopause is the coldest layer of the atmosphere (Zhao et al., 2020; Ortland et al., 1998). Polar summer mesopause is considered the coldest place on Earth (Ortland et al., 1998). The region has several unique physical and chemical characteristics including the complex interplay between radiative transfer, dynamics, and photochemistry (Smith, 2004). The height of the mesopause is not constant but varies significantly with latitude and season (Xu et al., 2007; Wang et al., 2022). The height is approximately 90 and 100 km in summer and winter respectively (Brasseur and Solomon, 2005). At mid and high latitudes the mesopause is located near 85km during the summer season (Smith, 2004). The mesopause of the high- and middle-latitude regions is at a lower and higher altitude in summer (around the summer solstice) and in winter/other seasons respectively. The mesopause is at a higher altitude at the equator for all seasons (Xu et al., 2007).

Spatial and temporal variability in temperature exists in the vertical temperature profile as well as changes with changing latitude. The temperature at the mesopause region exhibits a robust temporal variation (Mulligan et al., 1995; Offermann et al., 2010; Clancy and Rusch, 1989; Hedin, 1991; Dyrland et al., 2010; She et al., 2000; French et al., 2020b; Dalin et al., 2020; Grygalashvyly et al., 2014). Long-term and short-term temperature trends were studied by (Beig, 2011a, 2011b; Kalicinsky et al., 2016; Pancheva et al., 2013; She et al., 2015; Venkat Ratnam et al., 2010; Wörl et al., 2019; Xu et al., 2007; and recently reviewed by Gul 2024). Short-term variability in temperature is primarily due to small-scale gravity waves and tides (Dalin et al., 2017; Zhao et al., 2020). Air temperature measurements from the mesopause layer have long been important (Jarvis, 2001), because the cold temperatures of this layer are the potential tracers of the dynamics (Beig et al., 2003). The temperature during summer at the pole ranges between 120 to 140 K and at the winter pole range between 180 to 210 K (Brasseur and Solomon, 2005; Gul, 2024). This indicates that summer polar mesopause receives significantly more solar radiation as compared to winter mesopause, but the temperature is lowest at summer polar mesopause observed anywhere on Earth. The temperature response to solar activity is ~+2 times greater in winter than in summer (Dalin et al., 2020). Winter mesopause temperature trends (−6 to −2 K/decade) are generally stronger than summer ones (−2 to +0.5 K/decade (Offermann et al., 2010). Gravity and planetary





waves (Dalin et al., 2017), and atmospheric tides (Smith, 2004) bring periodic variations in
temperature.
WV is one of the strongest greenhouse gases in the atmosphere, important for cloud
formation, and plays a crucial radiative balance role in the atmosphere. WV in the upper
atmosphere can affect global surface climate (Solomon et al., 2010). WV in the atmosphere
regulates the Earth's weather and climate (Wallace and Hobbs, 2006). WV's existence in the
polar summer mesopause is of critical importance because it combines with the lowest
temperature of the mesopause region to enable noctilucent clouds (NLCs) to form. The
increased occurrence of NLC and its appearance is an indication of global change (Russell III
et al., 2014). Knowledge of WV distribution in the upper atmosphere is highly valuable for
understanding the respective roles of atmospheric chemistry, atmospheric dynamics, and
climate change. Complete methane ($CH_4$) oxidation is one of the major sources of WV
(Brasseur and Solomon, 1986). In the mesopause region, more abundant WV from increasing
$CH_4$ contributes to more frequent NLC occurrences (Lübken et al., 2018). The rocket
measurements gave water-mixing ratios of 3 to 7 ppm ranging between 40 and 70 km altitude
(Rogers et al., 1977). The airborne observation gave a water mixing ratio of 4 to 5 ppm at the
altitude range between 40 and 80 km (Waters et al., 1977). WV content in the atmosphere
controls the concentration of $O_3$ that, in turn, affects mesospheric cooling. The Photochemical
lifetime of $H_2O$ is relatively long making it an excellent tracer for atmospheric dynamics
(Peter, 1998) enabling one to follow the atmospheric transport effects up to high altitude
regions (80–85 km). The seasonal and long-term changes in mesospheric WV are discussed
by (Chandra et al., 1997). Due to the high sensitivity of WV to temperatures, NLC
phenomena can be used as temperature probes in the mesopause region (Lübken et al., 2007;
Petelina and Zasetsky, 2009) and as possible indicators of climate change (Thomas, 2003).
As compared to temperature, there have been fewer observations of WV in the upper
mesosphere.

96       2.   Methodology

2.1 Study area
Temporal and spatial variations of temperature and WV were monitored in the mesopause
region (Figure 1). Spatially the region was divided into four parts (North Pole, Equator, and
South Pole). Two two-degree latitude areas were selected for all longitude ranges (Figure 1).
Temporally twenty-two years (2002 - 2023) selected monthly data (as shown in Figure 1b)
was used from the TIMED SABER instrument. We used kinetic temperature (K) and $H_2O$



Mixing Ratio (ppmv) having dimensions of altitude, the event from SABER Custom Level2A
Product (Processed Level2A). We have excluded the four transitional months (November,
February, May, and August) and included four equinox months (March, April, September,
and October), and four solstice months (January, December, June, and July). We select
summer /northern hemisphere (NH) at ~80° ±1º latitude, equator at ~0° ±1º latitude, and 80°
±1º latitude in the winter/southern hemisphere (SH) for interannual variations of temperature
and WV during the study period. Temperature and WV trends are presented and compared
among different latitudinal and seasonal ranges. There is missing data in almost every
latitude (Zhu et al., 2005), particularly at high latitudes that vary with season. We applied a
weighted average to fill ~40% of missing values.
2.2 TIMED SABER instrument
SABER provides an excellent quality of the measured infrared limb radiances (Esplin et al.,
2023). SABER is an infrared limb-sounding instrument used for atmospheric sounding and
observing the atmosphere around the mesopause region continuously for over two decades.
Technical description of the SABER instrument and further relevant information are
discussed by Esplin et al., (2023); Mlynczak, (1997); and Russell III et al., (1999). Due to a
~60-day yaw cycle of the TIMED satellite, the latitude coverage shifts between 83°N–53°S
and 53°N–83°S. TIMED satellite rotates 180° about its yaw axis and provides latitude
coverage continuously in the range of 53°S to 83°N and then switching to 83°S to 53°N every
~60 days (Russell III et al., 1999). NASA-TIMED SABER instrument is performing near-
global measurements of the vertical kinetic temperature profiles along with volume mixing
ratios of WV. The random error of the v2.07 for SABER $H_2O$ product is 30% at 80km
altitude and further increases with increasing attitude (Rong et al., 2019). The rapid increase
in error is mainly due to low signal-to-noise. The estimated systematic error of SABER
version 2.07 $H_2O$ is about 10- 20%.



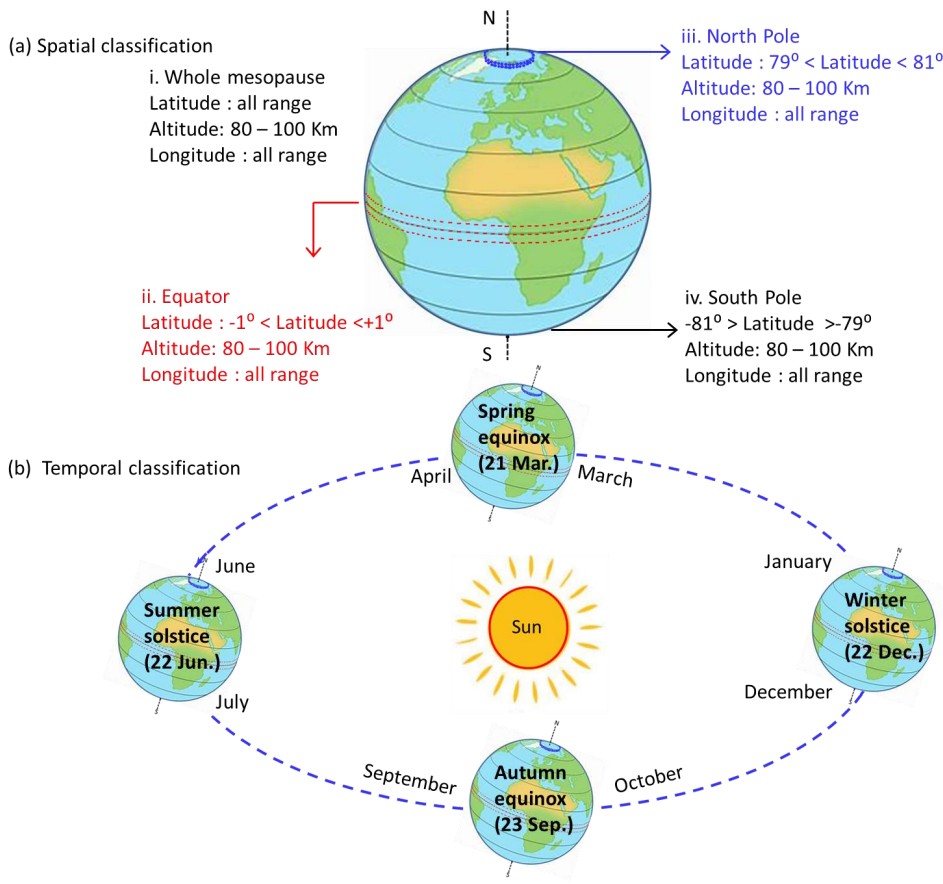


Figure 1. Study area (a) spatial range of selected regions, (b) temporal range from selected
years from 2002 to 2023.

3. Results

3.1. Variation in temperature and water vapor in the whole mesopause region

It is well known that temperature and WV in the atmosphere have high variability in space
and time. The vertical profiles of yearly averaged (selected 8 months averaged) temperature
and WV gradient with respect to mesopause altitude (km) are plotted in (Figure 2), described
in the following sections.

3.1.1 Variation in temperature

The year 2002 was the hottest (~193K) followed by 2003 (191K) and 2018 was the coldest
(~187K) year followed by 2008 (187.1K) during the study period Figure 2a. Yearly averaged
temperature decreased by ~6K from 2002 to 2008 and increased by ~3K from 2009 to 2012.



141 A second decrease in temperature ~4K was observed from 2014 to 2018. A decreasing trend

142 in temperature was observed ~0.37K decrease from 2002 to 2018 and ~0.14K decrease

143 during the whole study period 2002-2023. A decreasing trend in temperature (different

144 magnitude) was also reported by other authors in the past (Zhao et al., 2020; Dalin et al.,

145 2020; Yuan et al., 2019; Hervig and Siskind, 2006; French et al., 2020a; Venkat Ratnam et

146 al., 2010; Semenov et al., 2002). June 2008 (~180K) followed by July 2009 (~181K) were the

147 coldest months and January 2002 (~197K) was the hottest month during the study period.

148 March (max:196.8K min:188.3K avg:191.4K), April(max:195.2K min:189.6K avg:192K),

149 September(max:195.7K min:187.8K avg:190.6K), and October(max:195.8K min:189.3K

150 avg:191.7K) were relatively hotter months as compared to June and July. The monthly

151 averaged temperature at two equinoxes (Mar/Apr and Sep/Oct) were 191.66K and 191.16K

152 respectively. Similar monthly temperature patterns (lower temperatures during June and July)

153 were also reported in the past such as (Dalin et al., 2020; French et al., 2020; Offermann et

154 al., 2010). Therefore, seasonal temperature variations at the mesopause region are distinct,

155 with a summer minimum (June ~ 180K, July ~183K) and a winter maximum (January

156 ~197K, December ~190K). The monthly averaged temperatures at two solstices (Jun/July and

157 Dec/Jan) were 184.55K and 188.20K respectively, indicating the coldest temperature during

158 June and July throughout the whole study period. Mesopause during the summer (June and

159 July) solstice is ~ 3.65K colder than that during the winter (December and January) solstice

160 (~6–9 K colder was reported by Wang et al., (2022) and Xu et al., (2007). The seasonal

161 temperature variation is characterized by temporal variability in harmonics (Ammosov et al.,

162 2014; Kalicinsky et al., 2016; Perminov et al., 2014). Air is drawn downward in winter and

163 upward in summer keeping away mesopause from thermodynamic equilibrium. As a result,

164 the mesopause is kept away from thermodynamic equilibrium, with very low temperatures in

165 summer (June and July) and relatively high temperatures in winter (January and December).

166 The difference in temperature and the greater solar flux in December/January than in

167 June/July may also be due to the Earth's orbital eccentricity, as discussed by Chu et al.,

168 (2003).





Figure 2. Variation of temperature and water vapor in the mesopause region during 2002-2023 for the whole mesopause region.



Rising air expands and cools, resulting in a chilly summer mesopause, while downwelling air
compresses and warms, resulting in a warm winter mesopause. Downwelling in the winter
hemisphere and upwelling in the summer hemisphere causes adiabatic warming, and the
causes adiabatic cooling (States and Gardner, 2000; Xu et al., 2007). The transport of $CO_2$
affects the infrared cooling rate in the upper atmosphere leading to globally warmer
mesopause temperatures (Chabrillat et al., 2002) and enhanced differences in temperature
between the winter and summer. An analysis based on the work of (Dopplick, 1972; Kuhn
and London, 1969; López-Puertas et al., 1992), shows that the maximum cooling rate by $CO_2$
is found during the winter mesopause, where the temperature is relatively high. Thermal
infrared cooling is associated with WV, $CO_2$, and ozone and is a vital function of temperature
(Brasseur and Solomon, 2005). Mesospheric residual circulation is responsible for relatively
cold summers and warm winters. Temporal temperature variations may also be caused by
several factors including changes in the SOI-index, changes in the indices of geomagnetic
and solar activity (Medvedeva and Ratovsky, 2023), planetary, atmospheric, and meridional
circulation driven by breaking gravity waves (Offermann et al., 2009, 2011; Perminov et al.,
2014; Smith, 2012).
Temperature is also changing with respect to the change in altitude of the mesopause region.
Overall averaged temperature at 80km altitude was maximum (194.11K) and decreased by
~10K up to (184.72K) at the altitude of 97km as shown in Figure 2a. In general, lower
mesopause regions (80 – 90km) are relatively hotter than the upper part of mesopause (90-
100km). This temperature gradient was not consistent/similar for all selected months.
Different months showed slightly different patterns of temperature change with respect to
changing altitudes shown in Figure 2. Patterns of solstices (summer, and winter) were
different from the two equinoxes (Figure 2c, d, e). The temperature variation in June and
April (Figure 2c, d) are similar to the temperature variation in December (Figure 2e)
respectively.

3.1.2 Variation in water vapor
In the mesopause region, large fractional temporal and spatial variations in WV were
observed. Based on the monthly averaged WV for selected eight months of data, 2018 had a
relatively higher amount of water content (~1.14ppmv) followed by 2008 (1.14ppmv), and
2002 has the least amount of WV (~0.89ppmv) year followed by 2014 and 2003 (~1.0 ppmv)
during the study period Figure 2b. Overall an increasing trend in WV was observed



~0.13ppmv during the whole study period 2002-2023 (Figure 3a). An increasing trend in WV
was also reported by other authors in the past (Huaman and Balsley, 1999).WV content in the
mesopause region was in the range of ~ 0.05 ppmv(December 2009 at 100km altitude)–
4.81ppmv (June 2019 at 80km altitude) relatively smaller variation than previously reported
values ~0.1-10ppmv by other authors in the past (Berger and Von Zahn, 2002; Von Zahn and
Berger, 2003; Lubken et al., 2004; Lübken et al., 2009; Körner and Sonnemann, 2001;
Sonnemann et al., 2005). July 2008 (~1.48ppmv) followed by June 2019 (~1.45ppmv) were
the months had maximum WV content and April 2002 (~0.61ppmv) followed by October
2002 (~0.61ppmv) had minimum WV content during the study period. Monthly averaged
WV at two equinoxes (March/April and September/October) were 0.85ppmv and 0.88ppmv
respectively. In the mesosphere, the SABER $H_2O$ increasing trend was 0.1–0.2 ppmv per
decade (Yue et al., 2019), however, we observed a relatively lower increasing trend at the
equator (~0.09) and South Pole (~0.08) and North Pole (~0.06) ppmv/decade. An increasing
trend in WV in the lower atmosphere was also reported by (Oltmans and Hofmann, 1995;
Oltmans et al., 2000; Hurst et al., 2011; Nedoluha et al., 2013; Remsberg et al., 2018). The
Mesopause region showed a distinct pattern, with a summer maximum (June ~ 1.45ppmv,
July ~1.48ppmv) and a winter minimum (April ~0.61, October ~0.61ppmv). Monthly
averaged WV at two solstices (Jun/July and December/January) were 1.32ppmv and
1.23ppmv respectively. Nedoluha et al., (2022) showed maximum WV content during June
and July at 80km altitude. WV in the polar region is relatively higher in summer than in
winter. This may be due to upwelling in the summer hemisphere transports WV from lower
altitudes towards the mesopause (Körner and Sonnemann, 2001). There is no clear insitu
source of WV in the mesopause, except transported upwards from the stratosphere via the
meridional circulation and eddy transport, because of prevailing meridional circulation, and
Methane oxidation.
WV content is also changing with respect to the change in altitude of the mesopause region.
Overall averaged WV content at ~80km altitude was maximum (~3.16ppmv) and decreased
by ~3ppmv up to (~0.1ppmv) at the altitude of 97km as shown in Figure 2b. At 80 km
altitude, the WV mixing ratio ranges from ~ 1.5 to 4.5 ppmv reported by (Seele and Hartogh,
1999). In general, lower mesopause regions (80 – 83km) have relatively more WV content
than upper part of mesopause (84-100km). On average altitude above, 95km has very little
content of WV ~0.75ppmv. There are few studies including (Hervig et al., 2003) which
showed WV enhancement above 86 km altitudes. There is a distinct annual cycle of the WV



mixing ratio that can be seen in the three selected latitude ranges. The seasonal increase in
WV is relatively more prompt at lower altitudes of the mesopause region. The variations
(spatial and temporal) in atmospheric WV can be largely explained by dynamical factors
(quasi-biennial oscillation, the Brewer-Dobson circulation, and temperature changes)
(Dessler et al., 2014). The amount of WV in the region can also change with solar-cycle-
induced variations in Lyman-α radiation (Hervig and Siskind, 2006; Nedoluha et al., 2009).
On the solar cycle time scale, $H_2O$ may vary by about 30-40% near the mesopause height
(~80 km) caused by the solar cycle modulation of Lyman alpha (Chandra et al., 1997). At
mesospheric heights, WV is strongly photo-dissociated bysolar Lyman alpha (Brasseur and
Solomon, 1986). Solar cycle UV changes will have a strong influence on the long-term
changes in WV. Therefore, The solar cycle does play an important role in upper mesospheric
WV (Nedoluha et al., 2022). Additionally, changes in surface $CH_4$ emissions can increase the
amount of WV in the region due to $CH_4$ oxidation (Le Texier et al., 1988; Wrotny et al.,
2010). The secular increase in $H_2O$ related to methane increase in the atmosphere is about
0.4% /year at all heights in the mesosphere (Chandra et al., 1997). Model results suggest that
the temporal changes in mesosphere WV are largely controlled by the vertical advection
process associated with the meridional circulation (Chandra et al., 1997). The photolysis
process destroys $H_2O$ towards higher altitudes. The mesospheric WV content is the result of
the balance between its photodissociation and the upward transport from the stratosphere.
Temperature and WV content comparisons with past studies are shown in Table 1.

Table 1. Temperature and water vapor content comparisons with past studies in mesopause

| | Altitude(km) | Method / Location/Instrument | Reference |
|---|---|---|---|
| Temperature/trend | | | |
| 195K | 80 | TIMED SABER instrument | This study |
| 189K | 90 | TIMED SABER instrument | This study |
| 188K | 100 | TIMED SABER instrument | This study |
| -4K/decade | 80-100 | North Pole - TIMED SABER instrument | This study |
| -1.5K/decade | 80-100 | South Pole - TIMED SABER instrument | This study |
| -3K/decade | mesopause | LIMA model simulation | (Berger and Lübken, 2011) |
| -0.23K/year | 87 | GRIPS I and GRIPS II (small grating spectrometers of moderate resolution)/ Wuppertal (51°N, 7°E) | (Offermann et al., 2010), summer |
| -0.89K/year | mesopause region | Ground-based Infrared P-branch Spectrometer (GRIPS)/ Wuppertal (51° N, 7° E) | (Kalicinsky et al., 2016) |
| -2.5K/decade | 92-97 | The Na lidar at Fort Collins, CO (41°N, 105°W), and at Logan, UT (42°N, 112°W) | (Yuan et al., 2019) |
| -2.4K/decade | mesopause | OH* rotational temperature | (Dalin et al., 2020), summer |
| -0.4K/decade | mesopause | Moscow (Russia) ~57°N, 37°E | (Dalin et al., 2020), winter |
| -1.2K/decade | mesopause | LIMA and MIMAS model simulations 55–61°N | (Lübken et al., 2018) |
| -6.8K/decade | 100 | TIMED SABER instrument | (She et al., 2009) |
| -1.5K/decade | 91 | TIMED SABER instrument | (She et al., 2009) |




| | | | |
|---|---|---|---|
| -2.9K/decade | 80–105 km | Na lidar (41°N, 105°W) | (She and Krueger, 2004) |
| -2.1K/decade | mesopause | OH(6-2) rotational temperature (63°N) | (Ammosov et al., 2014) |
| -2K/decade | mesopause | OH* model simulations, | (Grygalashvyly et al., 2014) |
| -0.22K/year | mesopause | Airglow measurement | (Perminov et al., 2014) |
| -0.24K/decade | 80 - 88 km | Model simulations | (Hervig et al., 2015) |
| -0.5K/decade | mesopause | Model simulations | (Hervig et al., 2016) |
| -2K/decade | mesopause | Whole Atmosphere Community Climate Model eXtended (WACCM-X) | (Yuan et al., 2019) |
| -1.2K/decade | mesopause | OH nightglow rotational temperature | (French et al., 2020) |
| -0.3K/decade | mesopause | TIMED/SABER and airglow | (Noll et al., 2017) |
| Mixing ratio (ppmv) | | | |
| 3.65 | 80 | TIMED SABER instrument | This study |
| 0.88 | 90 | TIMED SABER instrument | This study |
| 0.07 | 100 | TIMED SABER instrument | This study |
| 0.06ppmv/decade | 80-100 | North Pole - TIMED SABER instrument | This study |
| 0.08ppmv/decade | 80-100 | South Pole - TIMED SABER instrument | This study |
| ~1.5 | 90 | SOFIE on AIM satellite and ALOMAR lidar | (Hervig et al., 2009)a |
| 1 | 90 | Using classical nucleation theory and a one-dimensional model | (Murray and Jensen, 2010) |
| 3 | 86 | ion-chemical model (A model that aims to describe these processes must combine basic ion chemistry, clustering processes as well as charge capture by particles) | (Gumbel et al., 2003) |
| 1 | 85 | Ground-based microwave techniques | (Bevilacqua et al., 1983) |
| 2.4 | 85 | Ground-based microwave technique at Norway (69ºN) | (Seele and Hartogh, 1999) |
| 3.4 | 84 | rocket-borne mass spectrometer (69°N, 16°E) | (Arnold & Krankowsky,1977) |
| 4 | 84 | space-borne methods/ SBUV (solar backscatter ultraviolet) satellite instrument, 77ºN | (Hervig et al., 2016) |
| 3 | 83 | 3-D model / upper mesosphere at high latitudes, 60ºN | (Von Zahn and Berger, 2003) |
| 4 | 83 | 3D-Model/mesopause / Smolarkiewicz scheme | (Körner & Sonnemann, 2001) |
| 1 | 83 | HALOE measurement / polar summer mesosphere | (Hervig et al., 2003) |
| 1 | 82 | 2-D numerical model | (Jensen and Thomas, 1994) |
| 1.8 | 80 | HALOE measurement /34ºN | (Russell III et al., 1993) |
| <4 | ~80 | HALOE measurement and chemical-dynamical model, | (Summers et al., 1997) |
| 0.45 to 4.81 | 80-94 | Model /Spitsbergen 78°N, | (Lubken et al., 2004) |
| 3 to 5.6 | 79-89 | Aura/MLS satellite /middle latitudes (45–50ºN) | (Dalin et al., 2023) |


### 3.1.3 Relationship between temperature and water vapor


Temperature showed a decreasing trend (discussed in section 3.1.1) and WV showed an


increasing trend (discussed in section 3.1.2) indicating a strong negative correlation between


WV and temperature Figure 3. A similar negative relation between WV with the solar cycle


at an altitude of 82km was also shown by Yue et al., (2019) and (Dalin et al., 2023). WV


measurements showed a decadal cycle (2002-2011, and 2011-2023) that is anticorrelated with


temperature variability, showing relatively more $H_2O$ during solar minimum Figure 3. A


similar anticorrelated relation between WV and solar variability was observed by (Hervig


and Siskind, 2006), and showed ~25% more $H_2O$ during solar minimum. Figure 3a shows the




yearly averaged temperature and WV content, where the opposition relationship is visible.
The year 2002 (maximum averaged temperature) and 2018 (minimum averaged temperature)
were the two extrema temperature years and were further explored on the monthly level in
part b (Temperature) and part c (WV) of Figure 3. The opposite correlation between
temperature and WV is true for a specific altitude. With increasing altitude (80 to 100km)
temperature and WV content both decreased (Figure 2). The precise relationship between
WV saturation mixing ratios and cold point temperature depends upon the temperature as
well as exact pressure (altitude), with Seidel et al., (2001) giving a value of ~0.6 ppmv/K,
(Fueglistaler and Haynes, 2005) ~0.5 ppmv/K, and (Nedoluha et al., 1998) ~0.7 ppmv/K. In
the present study, the maximum and minimum WV change between 81 to 100 km altitude
was ~4.3 and ~1.6 ppmv respectively. Solar cycle (temperature) variations impact on WV
and their relationship has been quantified by multiple researchers in the past  (Brasseur and
Solomon, 1986; Chandra et al., 1997; Fiedler et al., 2011; Hervig and Siskind, 2006; Siskind
et al., 2013).

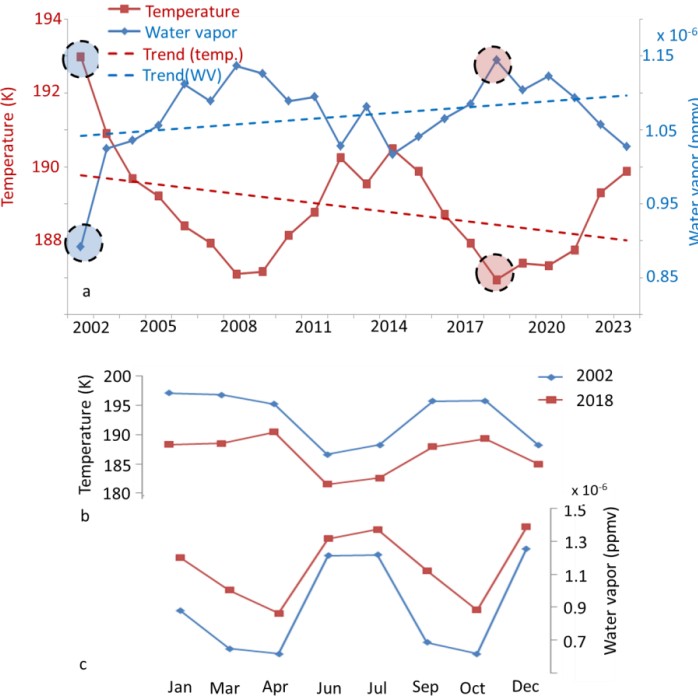


Figure 3. Relationship between temperature and water vapor content a. yearly averaged
temperature and water vapor in selected, b. temperature, c. water vapor for two selected
years.




3.2 Temperature and water vapor variation over the equator
Monthly averaged temperature data for selected eight months data, 2002 (~194K) was the hottest,
and 2019 (~188K) was the coldest year during the study period. June 2019 (~183K) was the coldest
and March 2014 (~198K) was the hottest month. Overall, there is a decreasing trend of temperature
(~1.5K/decade) over the equator. This decreasing trend in temperature is visible in all selected
months, particularly during September where this change is more prompt as compared to June and
July. The monthly averaged temperature at two equinoxes (Mar/Apr and Sep/Oct) were 194.15K and
192.07K respectively. Similarly, monthly averaged temperatures at two solstices (Jun/July and
Dec/Jan) were 185.81K and 188.95K respectively, indicating the coldest temperature during June and
July and the hottest temperature during March and April throughout the study period. Temperature
showed a decreasing trend with altitude in all selected months. Temperature decreased from 80 to 100
km during June and July was up to 10K, and during other months was around 20K.
Similarly, based on the monthly averaged WV data for selected eight months data, 2002 (~0.88ppmv)
followed by 2005 (~0.94ppmv) were the two least content WV years, and 2009 (~1.12ppmv) followed
by 2011 (~1.09ppmv) were the two maximum content of WV years during the study period. On
average July (1.25ppmv) had the maximum WV content followed by June (1.12ppmv) and October
(0.82ppmv) followed by April (0.87ppmv) had the minimum WV content. Overall, there is an
increasing trend of WV (~0.09ppmv/decade) over the equator of the mesopause region. The monthly
averaged WV at two equinoxes (Mar/Apr and Sep/Oct) was almost the same 0.90ppmv.
Similarly, monthly averaged WV at two solstices (Jun/July and Dec/Jan) were 1.19ppmv and
1.08ppmv respectively, indicating relatively high WV content during June and July. WV content at
lower altitudes 81, 82, and 83km were 2.97, 2.65, and 2.36ppmv respectively, and decreased with
altitude in all selected months. A decreasing trend in temperature and an increasing trend in WV is
clear in Figure 4a (yearly averaged). Monthly averaged variations in temperature and WV are shown
in Figure 4b, having a clear inverse relation in selected months and years Figure 4c.



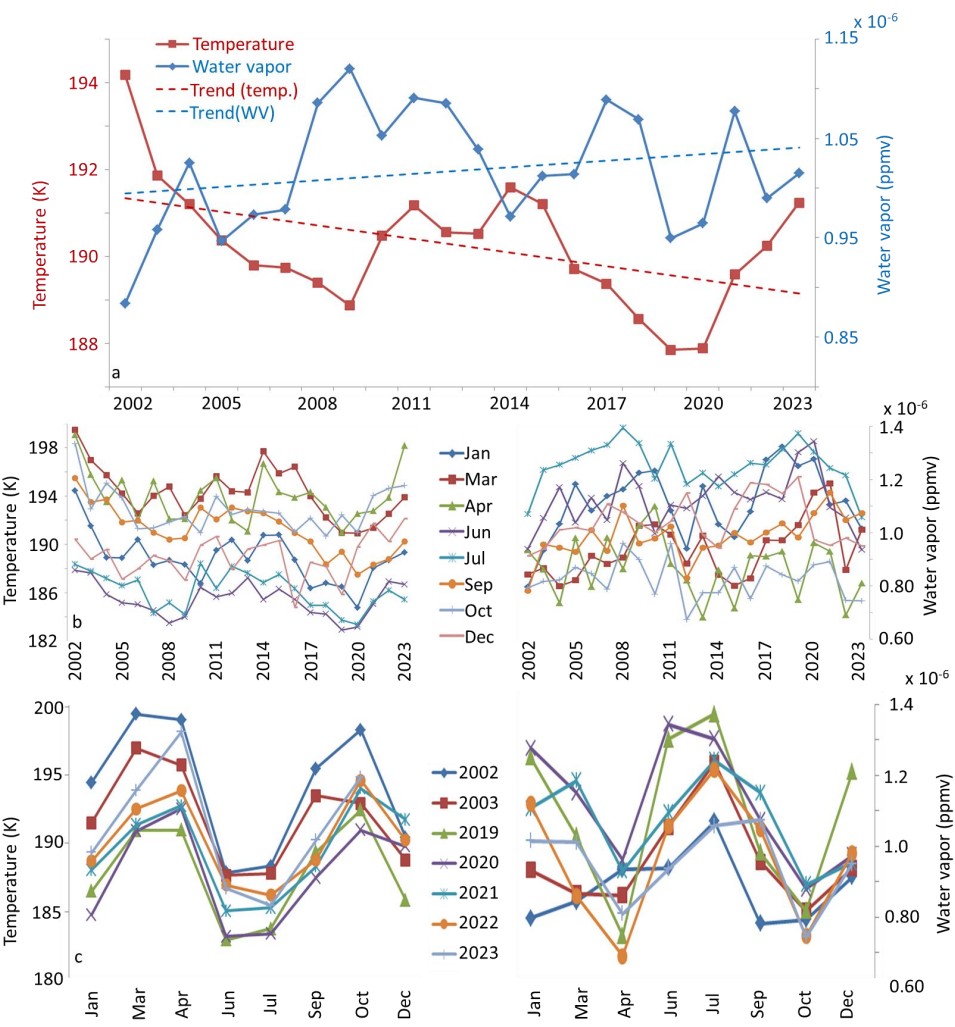

Figure 4. Temporal variation of temperature and water vapors over the equator
3.3 Temperature and water vapor variation over the North Pole
Monthly averaged temperature data for selected eight months data, 2002 (~190K) was the hottest, and
2023 (~182K) was the coldest year during the study period. June 2008 (~159K) was the coldest and
January 2002 (~210K) was the hottest month. Overall, there is a decreasing trend of temperature
(~4K/decade) in the North Pole of the mesopause region. This decreasing trend in temperature is more
prompt during September and less prompt during June and July. The monthly averaged temperature at
two equinoxes (Mar/Apr and Sep/Oct) were 185.57K and 191.50K respectively. Similarly, monthly
averaged temperatures at two solstices (Jun/July and Dec/Jan) were 162.64K and 201.14K
respectively, indicating the coldest temperature during June and July and the hottest temperature





during January and December throughout the whole study period. Our results are similar to those Xu
et al., (2007) showed a warmer mesopause at high latitudes during the December solstice than
it is in the June solstice. At high latitudes, the tendency term associated with vertical advection
reaches a maximum of 0.7 ppmv/day at 70-85 km in NH (Chandra et al., 1997). Generally,
temperature is not too much changing with altitude. The average temperature for all months
is around 200±10K in the mesopause region.  June showed a completely different pattern
than the other selected seven months. During June monthly temperature at 80km altitude was
(~160K) and further decreased up to ~134K at 90 km altitude and then sharply increased
(~250K) at the altitude of 100km. Average temperature during January was ~208±5K almost
constant with increasing altitude and showed very little decrease (~8K/20Km) altitude in
temperature. The temperature range during September was between 202-172K. There was a
clear temperature decrease between 84 to 96km during September. The temperature during
March showed a decreasing trend with increasing altitude. The average temperature at 80km
during March was around 210K and decreased to ~185K at an altitude of 100km.

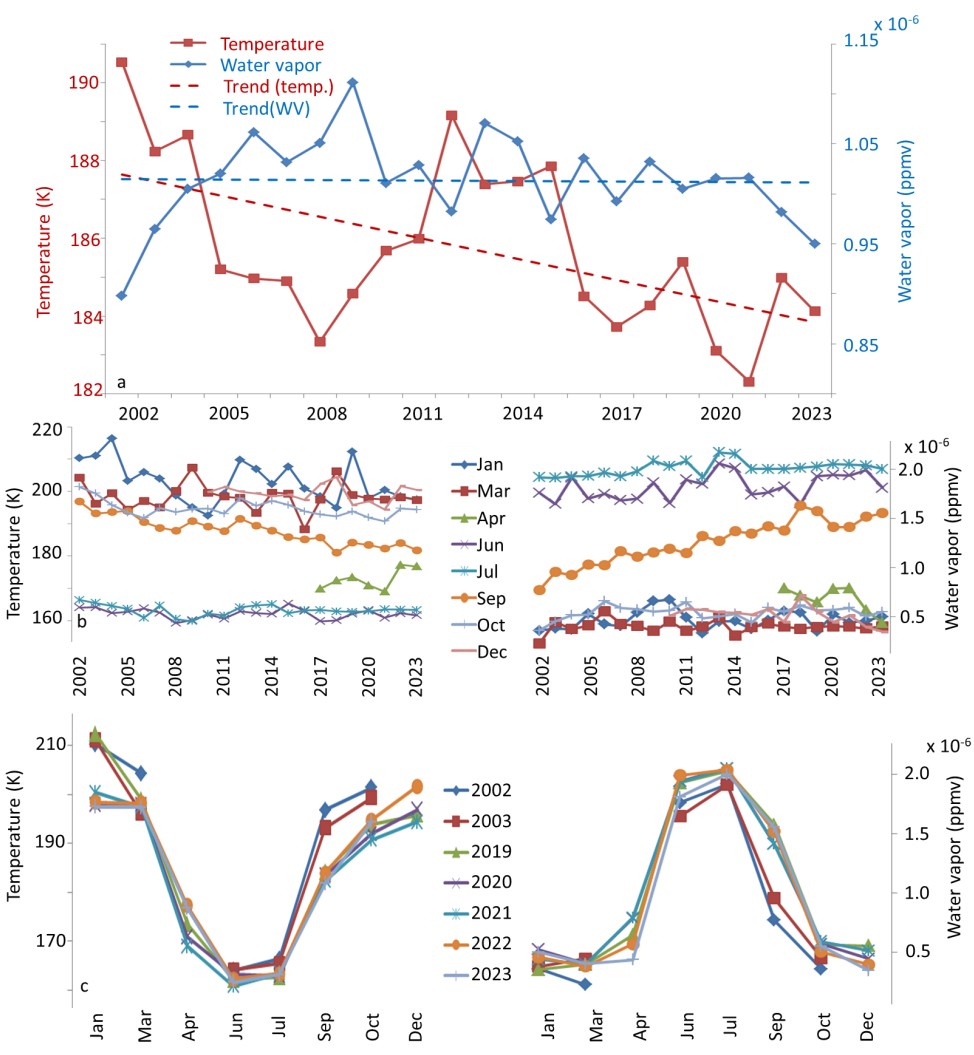


Figure 5 is the Same as Figure 4 but represents the North Pole
Monthly averaged WV data for selected eight months data, 2002 (~0.89ppmv) followed by 2023
(~0.95ppmv) were the two least content WV years, and 2009 (~1.11ppmv) followed by 2013
(~1.07ppmv) were the two maximum content of WV years during the study period (2002 – 2023). On
average July (~2.0ppmv) had the maximum WV content followed by June (~1.80ppmv) and March
(0.40ppmv) followed by January (0.47ppmv) had the minimum WV content. July 2013 (2.17)  and
March 2002 (0.22) were the maximum and minimum WV content months during the study period.
There is no clear increasing/decreasing trend of WV over the selected space of the North Pole of the
mesopause region. Monthly averaged WV at two equinoxes (Mar/Apr and Sep/Oct) was 0.54 and
0.90ppmv respectively.



Similarly monthly averaged temperature at two solstices (Jun/July and Dec/Jan) were 1.90ppmv and
0.49ppmv respectively, indicating relatively high WV content during June and July and low during
December and January, this is because at middle and high latitudes, the general transport of $H_2O$ is
directed upward in summer and downward in winter. Averaged WV content at lower altitudes 81, 82,
and 83km were 2.98, 2.73, and 2.40ppmv respectively, and decreased with altitude in all selected
months. As mentioned in section 2.2, SABER $H_2O$ is biased low by ~20% in polar summer above
80 km. It means SABER $H_2O$ reflects the polar winter and spring descent very well but in the
summer PMC region, the enhancement is weaker than expected. The decreasing trend in temperature
is clear but there is no clear trend in WV 5a (yearly averaged). Monthly averaged variations in
temperature and WV are shown in Figure 4b, having a clear inverse relation in selected months and
years Figure 5c.

3.4 Temperature and water vapor variation over the South Pole
Based on the monthly averaged temperature data for selected eight months data, 2002 (~188K) was
the hottest, and 2009 (~180K) was the coldest year during the study period. December 2007 (~156K)
was the coldest and March 2003 (~201K) was the hottest month. Overall, there is a decreasing trend
of temperature (~1.5K/decade) in the South Pole of the mesopause region. This decrease in
temperature was relatively clearly visible in March. On average April (197.64K) followed by June
and July were the hottest months and (December and January) were the coldest months throughout the
study period. Temperatures in January and December were 10 to 15K lower than the other six selected
months. The monthly averaged temperature at two equinoxes (Mar/Apr and Sep/Oct) were 194.85K
and 184.69K respectively. Similarly, monthly averaged temperatures at two solstices (Jun/July and
Dec/Jan) were 193.21K and 161.41K respectively, indicating the coldest temperature during January
and December and the hottest temperature during March and April throughout the whole study period.
Temperature variation with respect to altitude was different for January/December than for other
selected months. During winter (January/December) temperature was first decreased up to 92 Km
altitude and then increased to 93km and onward altitudes.

Monthly averaged WV data for selected eight months data, 2002 (~1.07ppmv) followed by 2012
(~1.08ppmv) were the two least content WV years, and 2009 (~1. 28ppmv) followed by 2008
(~1.28ppmv) were the two maximum content of WV years during the study period (2002 – 2023). On
average January (2.4 ppmv) had the maximum WV content followed by December (2.3 ppmv) and
September (0.45 ppmv) followed by April (0.47 ppmv) has the minimum WV content (Figure 6).
Overall, there is an increasing trend of WV (~0.08ppmv/decade) over the South Pole of the
mesopause region. The monthly averaged WV at two equinoxes (Mar/Apr and Sep/Oct) was 0.82 and
0.54ppmv lower than the monthly averaged temperature at two solstices (Jun/July and Dec/Jan) which





were 0.67ppmv and 2.3 ppmv respectively. This indicates relatively high WV content during winter
(December and January). In the SH temperature is colder at mid-to-high latitudes during January
(Wang et al., 2022), had relatively high WV content. At high latitudes, the tendency term associated
with vertical advection reaches a minimum of -0.35 ppmv/day in SH (Chandra et al., 1997). WV
content at lower altitudes 81, 82, and 83km were 3.25, 3.01, and 2.74ppmv respectively, and
decreased with altitude in all selected months. Temperature and WV trends and their interrelationship
are given in Figure 6.

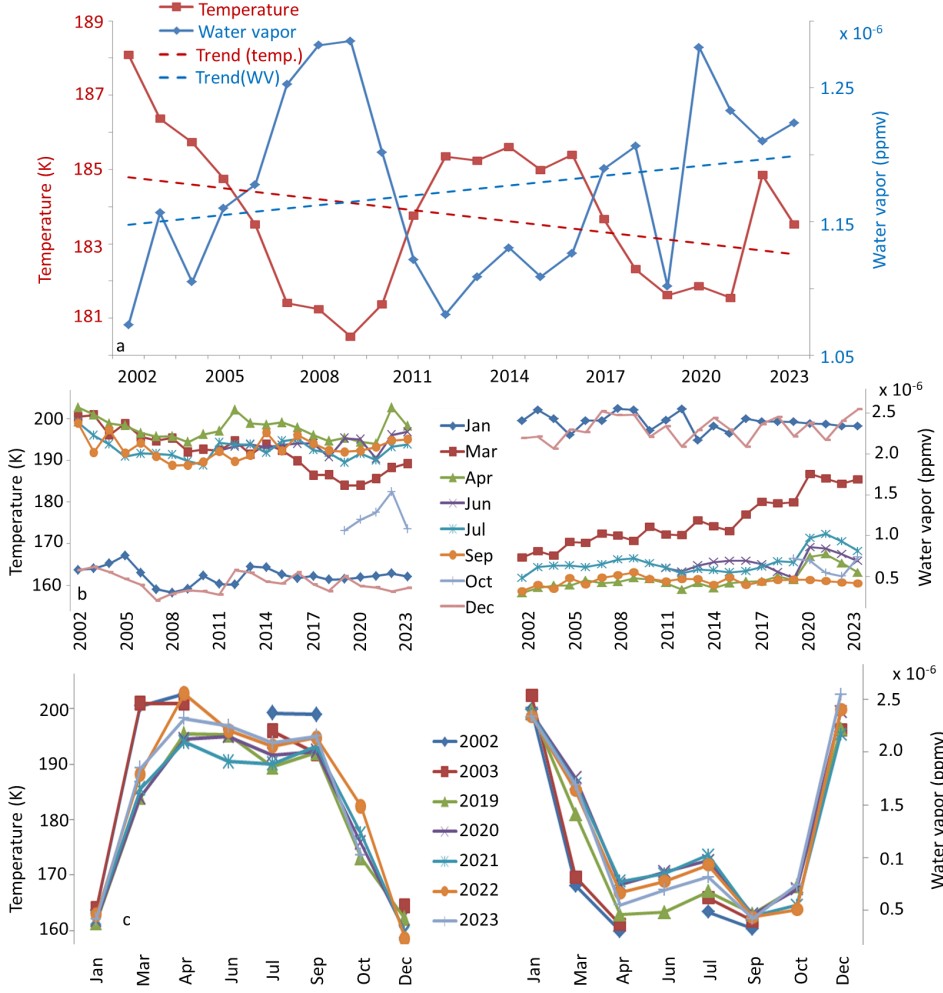


Figure 6. Same as Figure 4 but represents the South Pole region

395         4.   Inter and intra-comparison at solstices, equinoxes, and Poles

The average temperature at the summer solstice (June and July) is 162.64K, which is 38.5K lower
than that of the winter solstice (December and January). Similarly, the average WV at summer



solstice is 1.92ppmv which is ~1.43ppmv greater than that of winter solstice. The difference between
the Noth-South poles temperature and WV at equinoxes and soloists is shown in Figure 7.

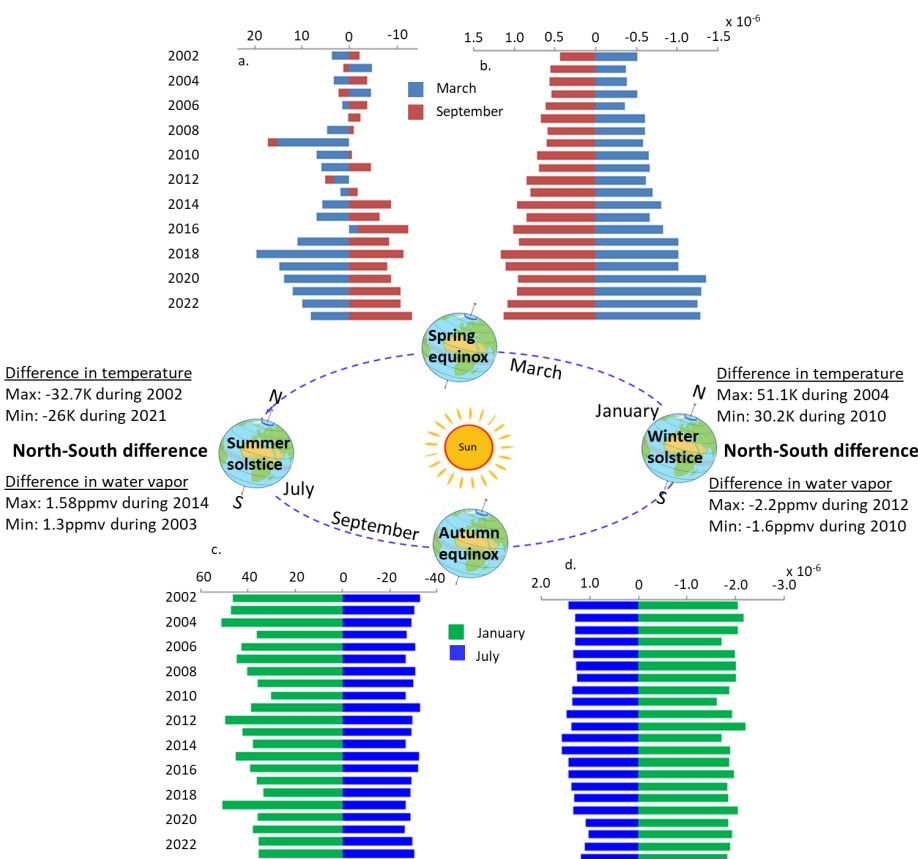


Figure 7. Difference in north-south temperature (a,c) and water vapor content (b,d) at two
equinoxes (March and September) and two solstices (January and July)
The difference between NH and SH temperature and WV at the solstice is higher than the difference
at the two equinoxes, indicating that the temperature and WV at both equinoxes are relatively close
however it look to increase in the future. The maximum and minimum difference in winter solstice
(January's) temperature (NH-SH) was during 2004 (51.11K) and 2010(30.24K) respectively, and in
summer solstice (July's) temperature (NH-SH) was during 2002 (-32.79K) and 2021(-26.60K)
respectively. Similarly, the maximum and minimum difference in July's WV (NH-SH) was during
2014(1.58ppmv) and 2003(1.3ppmv) respectively, and in January's WV (NH-SH) was during 2012 (-
2.2ppmv) and 2010(-1.60ppmv) respectively. For equinoxes WV and Temperature differences
between NH and SH are increasing with time, however, this difference for solstices is relatively
constant and has no increasing or decreasing trend with time.




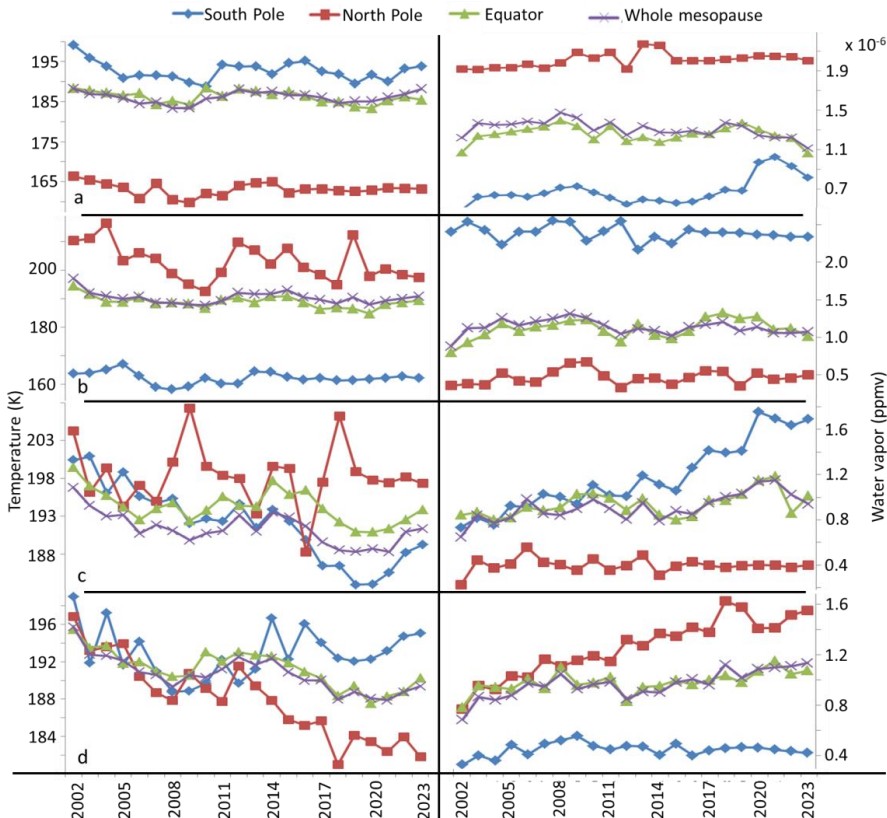

Figure 8. Intra-annual temperature and water vapor variations. The left column is for
Temperature and the right column is for water vapor in selected months a. July, b. January, c.
March, and d. September

The June and July temperature is colder at high latitudes (North Pole) in the summer hemisphere. In
addition, at summer high latitudes (North Pole for July and South Pole for January), the mesopause
temperature is ~5K colder. Examination of Figure 8 shows a clear temperature difference between
summer mesopause temperatures in the two hemispheres, with the southern temperatures appearing to
be at least a few K warmer during most of the summer season. The observed temperature differences
are highly significant ranging from 156K during December (South Pole), to around 210 K during
January (North Pole). The mean seasonal difference (the average of these values) is close to ~183K.
The inter-hemispheric comparison showed a clear difference in Temperature and WV. The difference
may be due to the reduced gravity waves and 50% weaker winds in the southern hemisphere
measured by Vincent, (1994), resulting in reduced mesospheric circulation. The summer season
ranges from May to August in the NH and from November to February in the SH, and is centered on
the solstice (Huaman and Balsley, 1999).



This result is close to (Wang et al., 2022)  and (Xu et al., 2007) who found that the mesopause during
the June solstice is colder than that during the December solstice. There is a slight difference in
magnitude may be due to the different lengths of the SABER temperature data set and using different
heights of the mesopause region. Xu et al., (2007) used initial 4 years of SABER data, and (Wang et
al., 2022) used 18 years of data set. Winter solstice (January) was the higher temperature month for
the North Pole and the lower temperature month South Pole (Figure 8). Similarly, the summer solstice
(July) was the higher temperature month for the South Pole and the lower temperature month North
Pole.  A similar but lower temperature difference was also observed for two equinoxes (Figure 8 left
column). A clear inverse relation between temperature and WV is shown in Figure 8.

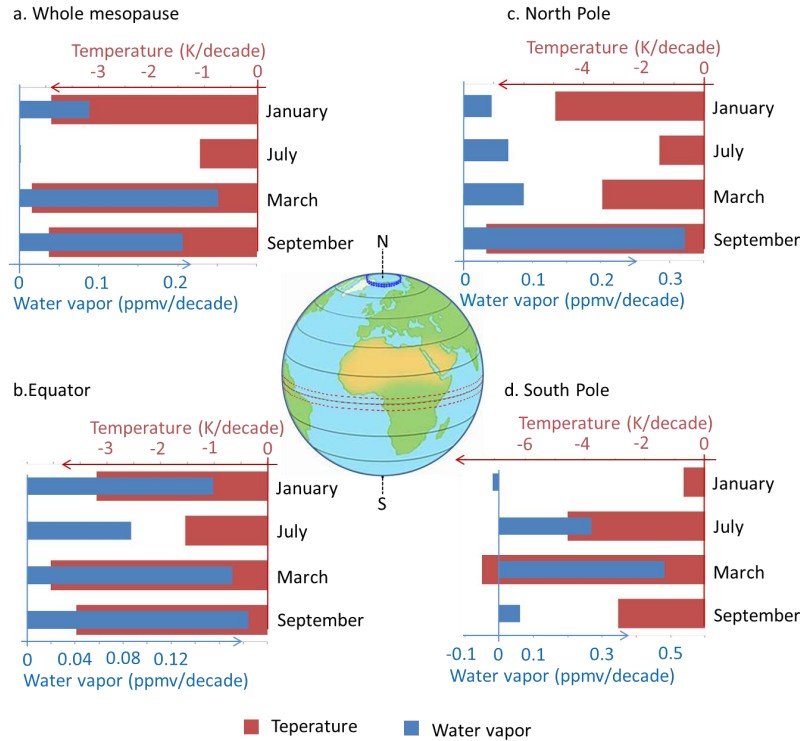


Figure 9. Temperature and water vapor trends for selected locations and months during the
study period.
At winter solstice (January), we observed less WV for the North Pole and relatively more WV for the
South Pole. Similarly, at the summer solstice (July) lower WV for the South Pole and more WV for
the North Pole. There is a relatively large difference in WV of winter solstice and summer solstices as
compared to the two equinoxes.
A decreasing trend in temperature and an increasing trend in WV in all selected locations (whole
mesopause, North Pole, South Pole, and the Equator) and selected months (January, July for solstices



and March, September for equinoxes) are shown in Figure 9. A cooling trend in temperature was also
reported in the past such as −9.2 K/decade in winter (Semenov, 2000), 0.64 K/decade (She et al.,
2015); −5.0K/decade (Winter) (Golitsyn et al., 1996), and −0.075 ± 0.043 K/year (Zhao et al., 2020).
Winter mesopause temperature trends (−6 to −2 K/decade) are generally stronger than summer ones
(−2 to +0.5 K/decade (Offermann et al., 2010). A maximum decreasing trend in temperature and an
increasing trend in WV is during March at the South Pole followed by September at the North Pole
(Figure 9). A slight increase in WV at the South Pole was observed during January. In the summer
hemisphere, the upwelling in high latitudes induces adiabatic cooling and creates a cold summer
mesopause at high latitudes (Wang et al., 2022). Upwelling causes a strong adiabatic cooling in the
summer mesosphere, affecting high latitudes. Therefore, at high latitudes in the summer hemisphere,
this adiabatic cooling effect (induced by the upwelling of the mesospheric circulation) creates a cooler
summer mesopause (Figure 9c). There is a strong upward transport by the mesospheric residual
circulation in the summer high latitudes. On the other side, in the winter hemisphere, the upwelling of
the lower thermosphere residual circulation causes cooling, and the downwelling of the mesospheric
residual circulation causes warming.

5.  Discussions
A variety of ground-based RADAR and LIDAR, satellite data, and model simulations (Table 1) have
been examined in the mesopause region. However, a few studies showed a detailed spatial and
temporal variation of temperature and WV. In this study, we presented the long-term differences in
temperature, and WV between the two hemispheres, two solstices, and two equinoxes in selected
months to gain a better understanding of spatial and temporal variation of temperature and WV at
high altitude mesopause.
The mesopause temperature during 2002–2023 showed a cooling trend through all selected latitudes
ranging from ~0.06 to -0.4 K/decade. Monthly averaged temperature data showed that 2002 was the
hottest and 2018 was the coldest year during the study period. A decreasing trend in temperature was
also reported by other authors in the past (Zhao et al., 2020 (Avg: −0.75K/decade); Dalin et al., 2020
(-2.4K/decade); Yuan et al., 2019 (~-2.4K/decade); French et al., 2020(-1.2K/decade)). Similarly,
Mlynczak et al., (2022) found significant cooling and contraction from 2002 to 2019 due to a weaker
solar cycle. According to Zhao et al., (2020), the cooling trends in the SH are stronger than those in
the NH. At the same time (Dalin et al., 2020) showed relatively stronger cooling at the summer
mesopause (−2.4K/decade), than that of winter mesopause (−0.4 K/decade). Our results showed a
significant cooling trend of ~4K/decade at the North Pole and a relatively low cooling trend
(~1.5K/decade) at the South Pole. The difference in results may be due to the difference in temporal
and spatial datasets. Seasonal temperature variations at the mesopause region are distinct, with a
summer minimum (June ~ 180K, July ~183K) and a winter maximum (January ~197K, December



~190K). The monthly averaged temperatures at two solstices (Jun/July and Dec/Jan) were 184.55K
and 188.20K respectively, indicating the coldest temperature during June and July throughout the
whole study period. Mesopause during the summer (June and July) solstice is ~ 3.65K colder than that
during the winter (December and January) solstice.
A clear hemispheric asymmetry in temperature (Figure 7) was observed, possibly related to solar
forcing and gravity wave forcing further discussed in (Xu et al., 2007). The mesopause temperature
varies from (Min: 182.3K, Max: 190.5K, Avg: 185.7K) in the summer polar region to (Min: 180.5K,
Max: 188.1K, Avg: 183.7K) in the winter polar region. Summer solstice was relatively colder
(~184K) than winter solstice (188K) for the whole mesopause case. Xu et al., (2007) showed a
warmer winter mesopause (~190K) than that of summer mesopause (~126K). In addition, at summer
high latitudes mesopause temperature (the North Pole for June/July) is 1.23K warmer than the South
Pole or December/January. At winter solstice (December and January), the North Pole's temperature
is higher than the South Pole's temperature and has relatively low WV content. At summer solstice
(June and July), the South Poles's temperature is higher than the North Pole Temperature. Wang et
al., (2022); Xu et al., (2007) found that the mesopause during the June solstice is ~6–9 K colder than
that during the December solstice. Huaman and Balsley, 1999 showed a predominant warmer SH (by
6K) around the summer solstice at 64⁰ latitude for a limited period (selected months of 1994),
measuring the temperature at different months in the two hemispheres. The slight difference in results
may be due to the different lengths of the SABER temperature data set and using different heights of
the mesopause region. We used a constant altitude of 80 to 100km during all seasons and latitudes can
change the results presented in this study than other studies in the past. The mesopause is ~1 km
higher at most latitudes and relatively warmer at middle to high latitudes in winter (around December
solstice) than it is in summer (around June solstice). At the equator, the mesopause is at a higher
altitude for all seasons. During the equinox months, the mesopause is at a constant altitude and
becomes discontinuous at the middle latitudes in the summer hemisphere from the equinoxes to the
solstices (Wang et al., 2022). Temporal and spatial variation of mesopause height is discussed by Xu
et al., (2007); Wang et al., (2022). The monthly averaged temperature at two equinoxes (Mar/Apr and
Sep/Oct) was 191.66K and 191.16K respectively indicating almost similar temperatures.
The adiabatic cooling effects are stronger due to stronger vertical wind at high latitudes and resulting
in a lower mesopause temperature and altitude (Wang et al., 2022). These adiabatic cooling effects are
strong enough to affect the mesopause altitude even during the transitional months between the
solstices and the equinoxes. Upwelling causes a strong adiabatic cooling in the summer mesosphere,
affecting high latitudes. Therefore, at high latitudes in the summer hemisphere, this adiabatic cooling
effect (induced by the upwelling of the mesospheric circulation) creates a cooler summer mesopause.
There is also a strong upward transport by the mesospheric residual circulation in the summer high
latitudes. On the other side, in the winter hemisphere, the upwelling of the lower thermosphere
residual circulation causes cooling, and the downwelling of the mesospheric residual circulation





causes warming. An increase in $CO_2$ was correlated with the decrease in temperature, which is one of
the possible reasons for the temperature-decreasing trend. $CO_2$ is transported from low altitudes and
affects the infrared cooling rate leading to globally warmer mesopause temperatures (Chabrillat et al.,
2002) and enhanced differences between the summer and winter temperatures (Smith, 2004). More
detailed analyses (of the energy budget, the response of nonmigrating tides, gravity waves,
geomagnetic activities, etc.) are required for further improvements.
WV measurements showed a decadal cycle (2002-2011, and 2011-2023) that was anti-correlated with
temperature variability and showed relatively more WV during solar minimum. A similar
anticorrelated relation between WV and solar variability was observed by (Hervig and Siskind, 2006),
and showed ~25% more WV during solar minimum. Anticorrelated relation between WV is also
shown by (Yue et al., 2019; Dalin et al., 2023). We find a maximum WV mixing ratio in June and
July and a higher annual variability than at the equator. At the polar summer mesopause, the WV
mixing ratio sharply rises from the beginning of May up to mid-September. The summer maximum
(North Pole) is a more isolated feature than at the equator: it has a rather sharp increase in WV in
April and a fast decline in autumn (September/October). Three-dimensional variations of temperature
and WV for January 2002 and July 2015 are shown in Figure 10.

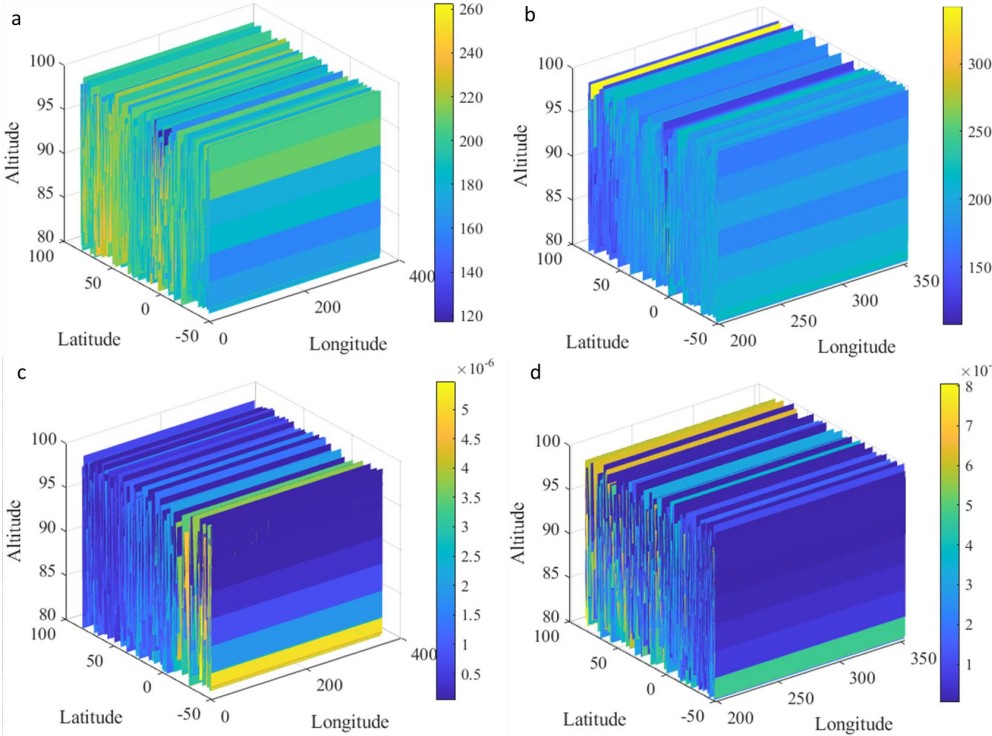


Figure 10. Three-dimensional variation of temperature and water vapor in selected months
a,c) January 2002 and b,d) July 2015 for the North Pole region.



A comparison of temperature and water vapor among the North Pole, South Pole, and Equator is
shown in Table 2.
Table 2. Comparison of temperature and water vapor among North Pole, South Pole, and
Equator

| | | North Pole | | South Pole | | Equator | |
|---|---|---|---|---|---|---|---|
| | | Maximum | Minimum | Maximum | Minimum | Maximum | Minimum |
| Temperature | Yearly | 2002 (190K) | 2023 (182K) | 2002(188K) | 2009(180K) | 2002(194K) | 2019(188K) |
| | Monthly | Jan -2002 (210K) | Jun-2008 (159K) | Mar-2003(201K) | Dec-2007(156K) | Mar-2014(198K) | Jun-2009(183K) |
| | Trend | Cooling trend ~4K/decade | | Cooling trend ~1.5K/decade | | Cooling trend ~1.5K/decade | |
| | Order | DJ(201K)>SO(191.5K)>MA(185.5K)>JJ(162.6K) | | MA(194K)>JJ(193K)>SO(184K)>DJ(161K) | | MA(194K)>SO(192K)>DJ(189K)>JJ(185K) | |
| | Altitude | 100km(282.7K) | 90km(125.3K) | 100km(261.1K) | 91km(134.1K) | 85km(215.6K) | 98km(171.9K) |
| Water vapor | Yearly | 2009(1.11ppmv) | 2002(0.89ppmv) | 2009(1.28ppmv) | 2002(1.07ppmv) | 2009(1.12ppmv) | 2002(0.88ppmv) |
| | Monthly | July-2015(2.0ppmv) | Mar-2023(0.4ppmv) | Jan-2016(2.4ppmv) | Apr-2002(0.30ppmv) | July-2008(1.39ppmv) | Oct-2023(0.74ppmv) |
| | Trend | Increasing trend ~0.06ppmv/decade | | Increasing trend ~0.08ppmv/decade | | Increasing trend ~0.09ppmv/decade | |
| | Order | JJ(1.90)>SO(0.9)>MA(0.54)>DJ(0.49) in ppmv | | DJ(2.3)>MA(0.82)>JJ(0.67)>SO(0.45) in ppmv | | JJ(1.19)>DJ(1.08)>MA(0.9)=SO(0.9) in ppmv | |
| | Altitude | 80km(7.18ppmv) | 100km(0.05ppmv) | 80km(7.34ppmv) | 100km(0.03ppmv) | 80km(4.14ppmv) | 100km(0.03ppmv) |

MA: March April averaged (spring equinox)
SO: September October averaged (autumn equinox)
JJ: June July averaged (summer solstice)
DJ: December January averaged (winter solstice)



### 6. Conclusions

In the present study, we used TIMED/SABER long-term monthly temperature and WV data to analyze the spatial and temporal distribution of the mesopause temperature and WV in selected timings and locations. The results indicate high spatial and temporal variation in temperature and had an inverse correlation with WV. Yearly averaged temperature for selected eight months of data, 2002 was the hottest (~193K) followed by 2003 (191K) and 2018 was the coldest (~187K) year followed by 2008 (187.1K) during the study period (2002 – 2023). Seasonal temperature variations at the mesopause region are distinct, with a summer minimum (June ~ 180K, July ~183K) and a winter maximum (January ~197K, December ~190K). The monthly averaged temperature at two solstices (Jun/July and December/January) were 184.55K and 188.20K respectively, indicating the coldest temperature during June and July throughout the whole study period. Mesopause during the summer (June and July) solstice is ~ 3.65K colder than that during the winter (December and January) solstice. A cooling trend was observed during twenty-two years of monthly observations. The cooling trends at high latitudes (North Pole) are relatively stronger than those at the Equator and South Pole. The mesopause temperature is colder in summer than in winter. Air is drawn downward in winter and upward in summer keeping away mesopause from thermodynamic equilibrium creating very low temperatures in summer (June and July) and relatively high temperatures in winter (January and December). The north-south temperature difference was maximum during the winter solstice of 2004. The vertical temperature gradient per km lies between -1.74 and 2.43 K during January, -2.14 and 1.28 K during March, -2 and 1.7 during April, -2 and 4.5 during June, -1.64 and 3.54 during July, -1.88 and 1.22 during September, -1.83 and 1.22K during October and -2.27 and 3.94K during December in the 80 to 100km altitude region, with values increasing in general with altitude and toward the summer pole (Schubert et al., 1990).

Based on the monthly averaged WV for selected eight months of data, 2018 had a relatively higher amount of water content (~1.14ppmv) followed by 2008 (1.14ppmv), and 2002 has the least amount of WV (~0.89ppmv) year followed by 2014 and 2003 (~1.0 ppmv) during the study period (2002 – 2023). July 2015 and January 2016 had maximum WV content at the summer and winter poles respectively. Overall there is an increasing WV trend in all selected locations. The north-south difference in WV was maximum during the winter solstice of 2012. The North Pole was coldest during the summer solstice and had a relatively large amount of water content. Upward transport, solar-cycle-induced variations in Lyman-α radiation, methane oxidation, and vertical advection are the responsible factors for changing WV content in the mesopause region. The possible loss of WV in the region is by photolysis and diffusion. We find that the occurrence of a WV maximum coincides with the temperature minimum in both hemispheres.



Author contributions
CG and SK initiated the idea; CG and DG performed the measurements and required calculations;
CG, DG and YY wrote the manuscript draft; SK and XG reviewed and edited the manuscript.
Competing interests
The authors declare that they have no conflict of interest.
Acknowledgments
The study was supported by the role of land air system in the arctic tropical correlation, National key
research and development plan (2022YFF0801703) and the State Key Laboratory of Cryosphere
Sciences (SKLCS20 ZZ-2022). We thank Faiza Gul for her encouragement and motivation. We
acknowledge NASA and supporting departments for the development of the SABER and TIMED
mission. Thanks to Global Atmospheric Technologies & Sciences (GATS) for providing data on
temperature and water vapor. We are grateful to the SABER retrieval team from GATS, Inc. for their
tireless effort to produce the SABER temperature and $H_2O$ data.

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
