# Peer review of "mesopause Region Chaman Gul\*1,2,3, Shichang Kang3, Yuanjian Yang1, Xinlei Ge2,4, Dong Guo\*2 1School of Atmospheric Physics, Nanjing University of Information Science & Technology, Nanjing, Jiangsu, 10044 China 2Reading Academy, Nanjing University of Information Science & Tech"

_EGUsphere, 2024_

## Referee Comment (RC1)

Review of "Spatial and temporal variation in long-term temperature and water vapor in the mesopause Region" by Gul et al.

This article investigates long-term change in temperature and water vapor using observations from the NASA SABER satellite instrument. The quality of writing, overall organization, and implementation of the English language are substandard, which detracts from the task of reviewing the scientific merit of the work. Regarding the scientific quality of the work, I believe there are major flaws that lead me to recommend rejecting this paper. It seems that poorly written papers are increasingly common, and I feel that the community is in danger of either lowering our standards or exhausting the review process.

Regarding the scientific quality of the paper, I have some important concerns which are the basis for my recommendation to reject this paper. Foremost is that there is no description of how the Authors determined trends from the observations. There are numerous resources that describe the derivation of trends from geophysical observations, and the Authors need to consider these methods and include the references. An important factor here is that the observed parameter is being modulated by another forcing mechanism, perhaps one that is periodic in nature, and that this dependence contaminates the derived trend. Of relevance here is that temperature and $H_2O$ in the mesosphere respond to the 11-yr solar cycle (see references in this paper), with less $H_2O$ and higher T near solar maximum. Looking at Figure 3, there is a clear 11-yr. solar cycle dependence in T and $H_2O$ (solar maxima were roughly 2002 and 2013). This is extremely important because the SABER time series begins near solar maximum and ends near solar minimum, giving the appearance of a massive cooling trend (and rising $H_2O$). If the Authors derived their trends from simple linear regression to the time series, then the results are likely not representative of the actual trends due to rising greenhouse gasses. The trends should be derived using multiple linear regression with the inclusion of at least two terms, 1) the solar cycle (e.g., using Lyman – alpha) and 2) time (i.e., the trend). The trend derived in this manner will be less affected by haphazard alignment between the observations and the solar cycle, as is clearly evident here. Many authors would choose to also include terms such as the QBO, AO, and ENSO. Again, there are accepted ways to do this and the Authors must adopt these approaches and describe what they did in the paper.

The paper suffers from a high degree of ambiguity in the presentation of their results, but also in the quotation of results from previous work. For example, look at the paragraph starting on line 470. The quoted trends are widely varying, yet there is no mention of the relevant latitude, altitude, or season for these results, and it is bewildering to try and make sense of it. This is just one example of the inadequate writing in this paper, and I feel that publishing these results in the present form would do more harm than good.

While this paper should be rejected, I believe that the subject matter is of interest and that it could represent a useful contribution after major revisions. To this end I offer some high-level suggestions below, but refrained from commenting on the ubiquitous flaws in writing, organization, and English, as this would consum too much time. I hope that the Authors will find some expert help to improve the writing and use of English. Please note that properly revising this paper will require *much* more consideration than offered in my comments below.

**Specific Comments**

**1) The writing, organization, and use of English are substandard.** To demonstrate this point, examine the first four sentences of the introduction:

> "Mesopause is one of the complex and intricate domain regions of Earth's atmosphere. It is the thermal transition area that plays an important role in the vertical coupling of the Earth's atmosphere. In the global mean temperature, mesopause is the coldest layer of the atmosphere (Zhao et al., 2020; Ortland et al., 1998). Polar summer mesopause is considered the coldest place on Earth (Ortland et al., 1998)."

The first sentence is confusing and serves no purpose. The second sentence is vague, and technically incorrect because much of the atmosphere is involved in some aspect of vertical coupling. The third and fourth sentences are awkwardly stated and somewhat redundant.

One more example from lines 99-100:

> "Spatially the region was divided into four parts (North Pole, Equator, and South Pole). Two two-degree latitude areas were selected for all longitude ranges (Figure 1)."

The first sentence states that the region (the globe?) was divided into 4 parts then mentions only 3, while the second sentence goes on to discuss only 2 latitude bands.

**2) Figure captions:** A figure caption must describe every aspect of the image, including details of the results (such as latitude, time, and height), and the origin of the results (such as "SABER observations" or "model results", or "trends derived from linear regression", etc…). It is not acceptable to do this only in the text, and then force the reader to go back and forth to understand a figure. An acceptable caption would be something like this (using Fig 4 as an example):

> **Figure 4.** Temperature and water vapor at 80 km altitude from SABER observations near the equator ($0° \pm 1°$ latitude). a) Time series of yearly mean T and $H_2O$, based on months as indicated in Figure 4b. Trends are also shown, which were determined using multiple linear regression to the results. b) T and $H_2O$ time series for individual months as indicated. c) T and $H_2O$ versus month for individual years as indicated.

**3) Figure 1:** This illustration is not needed, as most readers already understand these concepts. What would be much more useful is a plot of the SABER latitude coverage vs. time, as this is somewhat complicated. I show an example below of how this could be done. Note also that the SABER latitude vs. month is slowly changing over the years, and one must be very careful when constructing a 20+ yr. time series. For example, coverage of high northern latitudes included July in early years, but not during the recent several years. As a result, the Author's choice of June and July for high latitudes introduces a systematic bias in the time series, in that July is no longer represented in recent years. Illustrating these aspects of the data would be much more useful and relevant.

[Figure]

| | SABER latitude coverage versus time for observations during 2010. |

**4) Latitudes used in the study:** Given the excellent coverage provided by SABER, is there a reason to examine such narrow latitude bands (±1°), and only three latitudes (80°S, 0°, 80°N)? Regarding the ±1° latitude bands, I would generally expect a reduction in random variability for averaging over a wider latitude range (e.g., ±5°). The global mean (latitudes from 80°S - 80°N) is referred to as the "whole mesopause", which is ambiguous. Just call it the global mean. Furthermore, creating a global mean temperature can be misleading, since this approach combines different seasons in both hemispheres, and for SABER will include biases introduced by the changing latitude coverage with month (and year). For this reason, the best "global" representation of SABER data would be 52°S – 52°N, as has been done by previous authors.

**5) Altitudes used in the study:** It is not clear what altitudes were used for the T and $H_2O$ shown in the paper. I think it might be an average for 80-100 km (based on the legend in Fig 1), but it is not really stated clearly anywhere. This is an important point because the SABER errors increase rapidly with height. Furthermore, you should probably not mix measurements below and above the mesopause in a single average. In any case the paper should describe, and justify, the altitudes examined. Additionally, the Authors should consider looking at all altitudes.

**5) Table 1:** There appears to be a wealth of useful information here, but the results are poorly described and somewhat confusing. For example, the list appears to contain both absolute values of T and $H_2O$, in addition to trends in these quantities. The trends are also listed alternately as per year or per decade, and this needs to be rectified. Results are also given for a wide range of altitudes, which is problematic given the strong altitude dependence in T and $H_2O$ in the mesopause region. In addition, numerous investigations have shown that trends in the upper mesosphere vary strongly with height (references listed in this paper), and can even change sign at roughly NLC altitudes (depending on latitude and season). Given these complexities, the presentation of results in Table 1 needs to be substantially revised, including a consideration of the altitude dependence. Perhaps these results would lend themselves to being plotted vs. height instead. Finally, Table 1 neglects the $H_2O$ trends derived from SABER and MLS observations by Yue et al. (2022, GRL; reference given in this paper), which are particularly relevant to the present study.

**6) Figure 7:** These results are not described very well. What altitude is this for? Are you comparing Northern March to Southern March and Northern Sep. to Southern Sep.? If so then the differences are not meaningful as they are for two different seasons (e.g., spring vs. fall). Also, the diagram of Earth's orbital positions is not needed. Finally, a better illustration of these results would be a line plot of difference vs. year.

**7) Figure 8:** Why are you comparing the temperature and $H_2O$, for summer and winter, in different hemispheres? If your aim is to illustrate the seasonality then do it in the same hemisphere. If you are concerned with differences between hemispheres, then compare the same seasons in the north and south (e.g., for summer use June in the north vs. December in the south). Later in the text (line 421-423) you quote large seasonal differences of 156, 210, and 186K. Nowhere in the results are such differences evident, and this should be checked.

**8)** Throughout the article you refer to the "north pole" and "south pole", yet your results are for 80°N and 80°S, which are not the poles. Please be specific and use the nomenclature 80°N and 80°S.

**9) Figure 10:** The results are very hard to interpret, please try another approach.

---

## Referee Comment (RC3)

**Review of "Spatial and temporal variation in long-term temperature and water vapor in the mesopause Region" (egusphere-2024-1144) by Gul *et al*.**

General Comments

This paper presents an analysis of SABER temperature and water vapor profiles in the mesopause region (80-100 km). Selected months (solstices, equinoxes) and geographic regions (Equator, North and South polar) are examined throughout the 22-year data record covering 2002-2023. Extreme values are determined on annual and monthly basis for geographic and seasonal comparisons.

This paper provides relatively little new information. Numerous studies have examined the SABER data set previously. It is not clear that the authors have adequately addressed the changes in SABER sampling over its long data record. The trend analysis is too simplistic, and does not consider multiple periodic forcing functions that also affect the long-term variations of temperature and water vapor.

Definite need for grammar to be polished. Not trying to address all such items in this review.

Specific Comments

Page 1, lines 25-26: As discussed later, this statement does not take periodic forcing terms into account.

Page 3, lines 83-86: These are old references that only address a specific location and time, and discuss altitudes below the mesopause region. Why are they cited?

Page 3, lines 94-95: But there are certainly long data sets from ground-based microwave measurements and satellites (e.g. HALOE, SOFIE) that should be discussed.

Page 4, lines 111-112: 40% is a substantial amount of missing data. Does this represent frequent small gaps, or less frequent large gaps? What dimension is used for averaging? Altitude? Time? What is the weighting function?

Page 4, lines 120-122: This statement repeats the previous sentence. Since a 60-day yaw cycle is not an even fraction of a 365-day year, the latitude coverage in selected months will shift over 22 years. Are each of your months fully populated during the full data record?

Page 5, lines 133-135: Any comment on why the 2002 profile is such an outlier in the first 4 panels of Figure 2?

Pages 5-6, lines 138-143: There is a significant and well-known solar cycle signal in mesospheric temperature (noted on lines 242-249) that will greatly affect any calculated trends. Your results (which do not include any uncertainty estimate) cannot be compared to previous trends unless this contribution is addressed.

Page 6, lines 154-156: This statement seems very simplistic, given that your "global" average only includes small latitude bands near each pole and at the Equator. There is also significant altitude dependence that can vary between months and years.

Page 8, lines 201-204: The large variation in water vapor mixing ratio over this altitude range (as shown in Figures 2b and 2f) means that a simple average will be dominated by values from the lowest portion of the profile.

Pages 8-9, lines 204-206: Solar activity-induced variations will greatly affect any calculated trends in water vapor, as discussed previously for your temperature analysis.

Page 9, lines 224-225: Where is this result shown?

Pages 9-10, lines 237-238: But Figure 2 only shows global averages, not individual latitude ranges.

Page 10, lines 238-239: Again, Figure 2 only shows results averaged over 80-100 km, so what is the basis for this statement?

Page 10, lines 252-254: There are numerous studies during the last 30 years with more advanced models.

Page 10, line 257: Table 1 mixes different selections for latitude coverage, seasonal coverage, long-term temporal coverage, and data sources. It is hard to know what conclusions could (or should) be drawn.

Page 12, lines 280-283: This statement says that you have confirmed previous work. Any new information?

Page 15, lines 335-338: See previous questions about shift in yaw dates during SABER mission and impact on sampling. Note that the April data in Figure 5b only begin in 2017. Is the large trend in September temperature and water vapor affected by sampling changes?

Page 17, lines 350-354: Water vapor content at these low altitudes (81-83 km) will be affected by sublimation of PMC particles that settle from higher altitudes.

Page 17, lines 375-376: Note that January/December is summer in the Southern Hemisphere, not winter.

Page 27, line 586: It's not clear why you say "performed the measurements" when the paper only analyzes SABER data.

---

## Author Comment (AC1)

**Response to Reviewer 1 comments**
**Manuscript Number: EGUsphere-2024-1144**
**Manuscript title: Spatial and temporal variation in long-term temperature and water vapor in the mesopause Region, by Chaman Gul et al.,**

30th July 2024

Dear anonymous reviewer,

Thanks for the comments, suggestions, and recommendations for the EGUspher-2024-1144 manuscript. Comments are constructive and we quite improved the manuscript after addressing all the comments. We have thoroughly considered and carefully addressed all issues mentioned in the comments and have properly outlined every single change made in response to reviewer comments as suggested. We have made the required corrections in the revised manuscript (visible in tracked change mode) and prepared a list of point-by-point responses as given below starting from page #2 of this document. We have attached two copies of the revised manuscript, one with track change mode having all edits/corrections and the other is a fair copy of the manuscript where we have accepted all the mentioned edits/corrections. The reviewer's comments are in **black** text, the author's responses are in **blue** text, the modified/corrected text from the revised manuscript is in bold **brown** text, and references are in green text. Modified line numbers are in **yellow highlighted** text.

**Response to reviewer 1 (R1) comments (Cs):**

Reviewer #1: Review of "Spatial and temporal variation in long-term temperature and water vapor in the mesopause Region" by Gul et al.

**General comments by reviewer 1:**

**(R1-C1)** This article investigates long-term change in temperature and water vapor using observations from the NASA SABER satellite instrument. The quality of writing, overall organization, and implementation of the English language are substandard, which detracts from the task of reviewing the scientific merit of the work.

Response to (R1-C1):

Thank you very much for your precious time and multiple constructive comments. We have modified/revised the manuscript (including the language improvements), made the required corrections in the revised manuscript (visible in tracked change mode), and prepared a list of point-by-point responses as given below.

**(R1-C2)** Regarding the scientific quality of the work, I believe there are major flaws that lead me to recommend rejecting this paper. It seems that poorly written papers are increasingly common, and I feel that the community is in danger of either lowering our standards or exhausting the review process. Regarding the scientific quality of the paper, I have some important concerns which are the basis for my recommendation to reject this paper. Foremost is that there is no description of how the Authors determined trends from the observations. There are numerous resources that describe the derivation of trends from geophysical observations, and the Authors need to consider these methods and include the references. An important factor here is that the observed parameter is being modulated by another forcing mechanism, perhaps one that is periodic in nature, and that this dependence contaminates the derived trend. Of relevance here is that temperature and H2O in the mesosphere respond to the 11-yr solar cycle (see references in this paper), with less H2O and higher T near solar maximum. Looking at Figure 3, there is a clear 11-yr. solar cycle dependence in T and H2O (solar maxima were roughly 2002 and 2013). This is extremely important because the SABER time series begins near solar maximum and ends near solar minimum, giving the appearance of a massive cooling trend (and rising H2O). If the Authors derived their trends from simple linear regression to the time series, then the results are likely not representative of the actual trends due to rising greenhouse gasses. The trends should be derived using multiple linear regression with the inclusion of at least two terms, 1) the solar cycle (e.g., using Lyman – alpha) and 2) time (i.e., the trend). The trend derived in this manner will be less affected by haphazard alignment between the observations and the solar cycle, as is clearly evident here. Many authors would choose to also include terms such as the QBO, AO, and ENSO. Again, there are accepted ways to do this and the Authors must adopt these approaches and describe what they did in the paper.

Response to (R1-C2):

To investigate the long-term trend from the observations and solar response of the mesopause temperature, we used multiple linear regression analysis. We derived trends using multiple linear regression with the inclusion of two terms, 1) the solar cycle (e.g., using Lyman – alpha) and 2) time (i.e., the trend). We have updated the text accordingly (section 2.3.1 of the revised manuscript) and the same is given below

**"2.3.1. Multiple linear regression analysis**

**To investigate the long-term trends (temperature and WV) and the solar response of the mesopause temperature, a three-component harmonic fit is applied to remove the seasonality from the monthly data series. Then a multiple linear regression model is performed to solar activity, linear trend, and residual temperatures versus constant. Applying the regression analysis to latitude-averaged temperature and WV provides a more statistically significant value of their trends. Lyman-α flux is a proxy for solar activity, so the monthly mean of Lyman-α solar flux is used in multiple linear regression equation (1) as a measure of solar variability. Multiple linear regression analysis technique has been used by multiple authors in the past (e.g., Chandra et al., 1997; Hervig et al., 2015, 2016; Yue et al., 2019). To analyze the temperature and WV trends using multiple linear regression with the inclusion of the solar cycle and time we applied the following multiple regression analysis for trend estimation.**

$$\textbf{Temperature} = \textbf{C}_o + \textbf{C}_1(\textbf{Lyman}.\,\alpha) + \textbf{C}_2(\textbf{time}) + \textbf{error} \qquad (1)$$

**Where $C_0$ is constant (intercept), $C_1$ and $C_2$ are regression coefficients characterizing the linear long-term trend (temperature and WV per year) and solar activity term. We calculate temperature and WV trends using multiple linear regression involving monthly temperature and WV (SABER) data over time. Before applying the multiple regression model we calculate solar radiation according to monthly data sets. For example, Monthly means of the Lyman-α index are computed for each month, yielding 176 points for both global and equator."**

[Figure]

**A subpart of Figure 1:** Lyman- α index during January 2002-December 2023.

**(R1-C3)** The paper suffers from a high degree of ambiguity in the presentation of their results, but also in the quotation of results from previous work. For example, look at the paragraph starting on line 470. The quoted trends are widely varying, yet there is no mention of the relevant latitude, altitude, or season for these results, and it is bewildering to try and make sense of it. This is just one example of the inadequate writing in this paper, and I feel that publishing these results in the present form would do more harm than good.

Response to (R1-C3):

In the revised text we have provided detailed information as suggested. We have included the relevant latitude, altitude, or season for these results in the mentioned paragraph and other places in the manuscript. We have modified the paragraph as suggested (lines 613-617 of the revised manuscript) and the same is given below

**A cooling trend in temperature was also reported by other authors in the past (Zhao et al., 2020 (avg: −0.75 K/decade, latitude: 83°S to 83°N, altitude: 80-100 km); Dalin et al., 2020 (-2.4 K/decade, latitude: 57°N, altitude: 80-100 km, season: winter and summer); Yuan et al., 2019 (~-2.4 K/decade, latitude: ~42°N, altitude: 92 and 97 km, season: winter and summer ); French et al., 2020 (-1.2 K/decade, latitude: 68S, altitude: 87 km, season: winter)). Mlynczak et al. (2022) (latitude: 55°N to 55°S, altitude: mesosphere and lower thermosphere, season: annual) found significant cooling and contraction during 2002-2019 due to a weaker solar cycle.**

Information related to altitude, latitude, and season are also updated in the revised Table 1. Please have a look at the response to your comment (R1-C11, following pages) where we have provided the revised Table 1.

**(R1-C4)** While this paper should be rejected, I believe that the subject matter is of interest and that it could represent a useful contribution after major revisions. To this end I offer some high-level suggestions below, but refrained from commenting on the ubiquitous flaws in writing, organization, and English, as this would consum too much time. I hope that the Authors will find some expert help to improve the writing and use of English. Please note that properly revising this paper will require much more consideration than offered in my comments below.

Response to (R1-C4):

Thank you for allowing us to improve the writing and use of English. The paper has been carefully reviewed by the co-authors. Multiple edits/improvements related to writing, organization, and English are visible in the revised track-changed version of the manuscript. A few examples of these updates are given below.

1. We revised all figures and tables.
2. We added additional required sections, for example, section 2.3 Solar cycle response, and section 2.3.1 Multiple linear regression analysis.
3. We removed less relevant (or having a repetition of information) sections, for example, section 3.1.3 (relationship between temperature and WV). We already showed an inverse relation between temperature and WV in multiple locations.
4. We rewrite whole sections, for example first section of the introduction part.
5. We revised the captions of all figures as suggested
6. We have made corrections in almost every line of the revised manuscript.

 **Specific comments by reviewer 1:**

**(R1-C5) 1) The writing, organization, and use of English are substandard**. To demonstrate this point, examine the first four sentences of the introduction:

> "Mesopause is one of the complex and intricate domain regions of Earth's atmosphere. It is the thermal transition area that plays an important role in the vertical coupling of the Earth's atmosphere. In the global mean temperature, mesopause is the coldest layer of the atmosphere (Zhao et al., 2020; Ortland et al., 1998). Polar summer mesopause is considered the coldest place on Earth (Ortland et al., 1998)."

The first sentence is confusing and serves no purpose. The second sentence is vague, and technically incorrect because much of the atmosphere is involved in some aspect of vertical coupling. The third and fourth sentences are awkwardly stated and somewhat redundant.

Response to (R1-C5):

We tried our best to improve writing, organization, and the use of language to present a better quality of work. We have revised/updated the whole paragraph (lines 47-69 of the revised manuscript) and the same is given below.

In the global mean temperature, mesopause is the coldest layer of the atmosphere (Zhao et al., 2020; Ortland et al., 1998), and exhibits a robust variation in temperature (Offermann et al., 2010; Dyrland et al., 2010; She et al., 2000; French et al., 2020b; Dalin et al., 2020; Grygalashvyly et al., 2014). Temperature is changing from ~160 to ~185 K, relatively cooler over the equatorial region and warmer toward both poles (Xu et al., 2007). The temperature at the summer pole ranges between 120 to 140 K and at the winter pole ranges between 180 to 210 K (Brasseur and Solomon, 2005). The amplitude of seasonal variations in the mesopause temperature increases with increasing latitude. The temperature and mean location of menopause are established by radiative and dynamical processes (e.g., Leovy, 1964; Holton, 1983) and also display large variability due to tides, gravity, and planetary waves. Short-term variability in temperature is primarily due to small-scale gravity waves and tides (Dalin et al., 2017; Zhao et al., 2020). Gravity and planetary waves (Dalin et al., 2017), and atmospheric tides (Smith, 2004) bring periodic variations in temperature. The temperature response to solar activity is ~+2 times greater in winter than in summer (Dalin et al., 2020). Winter mesopause temperature trends (−6 to −2 K/decade) are generally stronger than summer ones (−2 to +0.5 K/decade) (Offermann et al., 2010). This indicates that summer polar mesopause receives significantly more solar radiation than winter mesopause, but the temperature is lowest at summer polar mesopause observed anywhere on Earth. The height of the mesopause varies significantly with latitude and season (Xu et al., 2007; Wang et al., 2022). The mesopause height is approximately 90 and 100 km in summer and winter respectively (Brasseur and Solomon, 2005). At mid and high latitudes the mesopause is located near 85 km during the summer season (Smith, 2004). The mesopause is at a higher altitude at the equator for all seasons (Xu et al., 2007).

In the revised manuscript, the introduction section has four paragraphs. The first paragraph is about mesopause temperature and altitude. The second paragraph is about water vapor in the mesopause/atmosphere and its connection with temperature. The last paragraph has information about current work. Besides above mentioned paragraph, we made similar changes in other multiple places of the manuscript.

(R1-C6) One more example from lines 99-100:
"Spatially the region was divided into four parts (North Pole, Equator, and South Pole). Two two-degree latitude areas were selected for all longitude ranges (Figure 1)."

The first sentence states that the region (the globe?) was divided into 4 parts then mentions only 3, while the second sentence goes on to discuss only 2 latitude bands.

Response to (R1-C6):

We have revised the whole section, and clearly explained the study area (lines 110-124 of the revised manuscript), and the same is given below

**"2.1. Study area**

**Spatial and temporal variations in long-term temperature and WV are analyzed in the mesopause region (80-100 km altitude). Spatially the region is divided into three latitude bins (Equator, Northern, and Southern Hemispheres). Taking temperature and WV data in $0° ± 1^o$, $80°N ± 1^o$, and $80°S ± 1^o$ latitude bins represent the equator, northern hemisphere (NH), and southern hemisphere (SH) respectively. All latitudes, and longitudes of the mesopause covered by the SABER instrument during a year are represented by global mesopause. Twenty-two years (2002-2023) of monthly data from the SABER instrument on board the NASA Thermosphere, Inonosphere, Mesosphere, Energetic, and Dynamics (TIMED) satellite are analyzed during eight selected months of each year excluding the four transitional months (February, May, August, and November). The selected months for temporal analysis of temperature and WV are four equinoxes (March, April, September, and October), and four solstices (January, December, June, and July) months……….."**

**(R1-C7) Figure captions:** A figure caption must describe every aspect of the image, including details of the results (such as latitude, time, and height), and the origin of the results (such as "SABER observations" or "model results", or "trends derived from linear regression", etc…). It is not acceptable to do this only in the text, and then force the reader to go back and forth to understand a figure. An acceptable caption would be something like this (using Fig 4 as an example): **Figure 4**. Temperature and water vapor at 80 km altitude from SABER observations near the equator ($0° ± 1°$ latitude). a) Time series of yearly mean T and H2O, based on months as indicated in Figure 4b. Trends are also shown, which were determined using multiple linear regression to the results. b) T and H2O time series for individual months as indicated. c) T and H2O versus month for individual years as indicated.

Response to (R1-C7):

We have revised all Figure's captions as suggested. A comparison table of previously used captions and revised captions along with page numbers in the revised text is given below

| Fig. # | Previous caption | Revised caption used in the revised manuscript | Page # |
|---|---|---|---|
| 1 | Study area (a) spatial range of selected regions, (b) temporal range from selected years from 2002 to 2023. | SABER instrument latitude coverage versus time for observation and Lyman-α solar index. a) Monthly data coverage in selected months versus latitude ranges from January 2002 to December 2023, excluding transitional months. b) Comparison of SABER latitude coverage and monthly data versus time during years (2002-2003). c) Typical temporal coverage of TIMED-SABER instrument measurements. d) Latitude versus longitude tangent point locations for one day of observations in its north viewing phase (83°N to 52°S) – a north viewing yaw mode. e) Lyman- α index during January 2002-December 2023. | 7 |
| 2 | Variation of temperature and water vapor in the mesopause region during 2002- 2023 for the whole mesopause region. | Temperature and water vapor gradient between 80-100 km altitudes from SABER observations at the three selected latitude bins during 200-2023. a) Equator ($0° \pm 1°$). b) Northern hemisphere ($80°N \pm 1°$). (c) Southern hemisphere ($80°S \pm 1°$), in the indicated months, by averaging all January, June, and September values from 2002 to 2023. | 11 |
| 3 | Relationship between temperature and water vapor content a. yearly averaged temperature and water vapor in selected, b. temperature, c. water vapor for two selected years. | Temperature and water vapor at 80-100 km altitude from SABER observations on the global scale. a) Time series of yearly mean temperature and WV, based on selected months as indicated in Figure 3c. b) Differences of the SABER annual temperature and fit curve (residuals). c) Temperature and WV time series for individual months of 2002 and 2018. d) Differences in the SABER annual temperature and fit curve during 2002. | 16 |
| 4 | Temporal variation of temperature and water vapors over the equator | Temperature and water vapor at 80-100 km altitude from SABER observations near the equator ($0° \pm 1°$ latitude). a) Time series of yearly mean temperature and WV, based on months as indicated in Figure 4c. b) Differences of the SABER annual temperature and fit curve (residuals). c) Temperature and WV time series for individual months as indicated. d) Temperature and WV versus month for individual years as indicated. | 19 |
| 5 | is the Same as Figure 4 but represents the North Pole | Temperature and water vapor at 80-100 km altitude from SABER observations near the NH | 21 |

| | | (80° ± 1° latitude). a) Time series of yearly mean temperature and WV, based on months as indicated in Figure 5c. b) Differences of the SABER annual temperature and fit curve (residuals). c) Temperature and WV time series for individual months as indicated. d) Temperature and WV versus month for individual years as indicated. | |
|---|---|---|---|
| 6 | Same as Figure 4 but represents the South Pole region | Temperature and water vapor at 80-100 km altitude from SABER observations near the SH (80°S ± 1° latitude). a) Time series of yearly mean temperature and WV, based on months as indicated in Figure 6c. b) Differences of the SABER annual temperature and fit curve (residuals). c) Temperature and WV time series for individual months as indicated. d) Temperature and WV versus month for individual years as indicated. | 24 |
| 7 | Difference in north-south temperature (a,c) and water vapor content (b,d) at two equinoxes (March and September) and two solstices (January and July) | Difference in NH-SH temperatures and NH-SH, WV at 80-100 km altitude during selected months of equinoxes (March and September) and solstices (January and July). a, c) Water vapor content difference. b, d) Temperature difference. | 26 |
| 8 | Intra-annual temperature and water vapor variations. The left column is for Temperature and the right column is for water vapor in selected months a. July, b. January, c. March, and d. September | Inter-annual variations in monthly mean temperature and water vapor from SABER observations over selected bins of latitudes during 2002-2023. The left column is for temperature temporal variation and the right column is for water vapor temporal variation in selected months. a) 22 years monthly mean for July. b) 22 years monthly mean for January. c) 22 years monthly mean for March, and d) 22 years monthly mean for September. | 27 |
| 9 | Temperature and water vapor trends for selected locations and months during the study period. | Temperature and water vapor trends during selected four months as indicated. a) Temperature and WV trends over NH (80ºN ± 1º). b) Temperature and WV trends over SH (80ºS ± 1º). c) Temperature and WV trends over the equator (0º ± 1º). d) Temperature and WV trends on the global scale. | 29 |
| 10 | Three-dimensional variation of temperature and water vapor in selected months a,c) January 2002 and b,d) July 2015 for the North Pole region. | Two-dimensional (latitude, and altitude) variation in temperature (K), and WV (ppmv) at three latitudes as indicated, and three altitudes (80 km, 90 km, and 100 km) during January 2003. | 34 |
| 11 | This is a new added figure in the revised manuscript. | Two-dimensional (latitude, and altitude) variation in temperature (K), and WV (ppmv) at three latitudes as indicated, and three altitudes (80 km, 90 km, and 100 km) during June 2023. | 35 |

**(R1-C8) Figure 1**: This illustration is not needed, as most readers already understand these concepts. What would be much more useful is a plot of the SABER latitude coverage vs. time, as this is somewhat complicated. I show an example below of how this could be done. Note also that the SABER latitude vs. month is slowly changing over the years, and one must be very careful when constructing a 20+ yr. time series. For example, coverage of high northern latitudes included July in early years, but not during the recent several years. As a result, the Author's choice of June and July for high latitudes introduces a systematic bias in the time series, in that July is no longer represented in recent years. Illustrating these aspects of the data would be much more useful and relevant.

Response to (R1-C8):

We have replaced Figure 1 with a new Figure as suggested (page 7 of the revised manuscript) and the same figure is given below.

[Figure]

**Revised Figure 1**. SABER instrument latitude coverage versus time for observation. a) Monthly data coverage in selected months versus latitude ranges from January 2002 to December 2023, excluding transitional months. b) Comparison of SABER latitude coverage and monthly data versus time during years (2002-2003). c) Typical temporal coverage of TIMED-SABER instrument measurements. d) Latitude versus longitude tangent point locations for one day of observations in its north viewing phase (83°N to 52°S) – a north viewing yaw mode.

We have illustrated the mentioned aspects in the methodology section (lines 126-164 of the revised manuscript) and the same is given below

**"2.2. TIMED-SABER instrument**

**The TIMED-SABER satellite views 90° to the right of the velocity vector of the TIMED spacecraft, and completes a full 24-hour local time coverage in 60-63 days (Russell III**

et al., 1999; Mlynczak et al., 2003; Figure 1). The SABER instrument scans the atmosphere from the troposphere up to the lower thermosphere and obtains vertical profiles kinetic temperature and volume mixing ratio of WV (Russell et al., 1999). The instrument performs near-global measurements and provides an excellent quality of the measured infrared limb radiances (Esplin et al., 2023). Technical description of the SABER instrument and further relevant information are discussed by Mlynczak, (1997) and Russell III et al. (1999). TIMED satellite rotates 180° about its yaw axis and provides latitude coverage continuously in the range of 53°S to 83°N and then switching to 83°S to 53°N every ~60 days (Russell III et al., 1999). Due to the asymmetrical latitudinal coverage of the SABER instrument, there are some missing measurement months at high latitudes (52°N-83°N or 52°S-83°S). Multiple studies ( e.g; Forbes et al., 2021; Liu et al., 2017; Das, 2021) are limited to the latitude band ~50°S to ~50°N, mainly due to the TIMED ~60 days yaw cycle. In the present study, we have included high-latitude regions from both hemispheres along with some missing data. For example, coverage of high northern latitudes included July in the early years, but not during the recent several years (2017-2023)………" Please have a look at this section in the revised manuscript for full details.

(R1-C9) Latitudes used in the study: Given the excellent coverage provided by SABER, is there a reason to examine such narrow latitude bands (±1°), and only three latitudes (80°S, 0°, 80°N)? Regarding the ±1° latitude bands, I would generally expect a reduction in random variability for averaging over a wider latitude range (e.g., ±5°). The global mean (latitudes from 80°S - 80°N) is referred to as the "whole mesopause", which is ambiguous. Just call it the global mean. Furthermore, creating a global mean temperature can be misleading, since this approach combines different seasons in both hemispheres, and for SABER will include biases introduced by the changing latitude coverage with month (and year). For this reason, the best "global" representation of SABER data would be 52°S – 52°N, as has been done by previous authors.

Response to (R1- C9):

A relatively wider latitude range is multime used in the past. For example, a 10° latitudinal band from 83°S to 83°N is recently used by Zhao et al. (2020). Wang et al. (2022) investigate the seasonal variability of the residual circulations and the mesopause temperature at different latitudes by selecting four 20° latitudinal bands centered at 10°S and 10°N for low latitudes, and 50°S and 50°N for mid-to-high latitudes. Grygalashvyly et al. (2014) subdivided the

latitudes into 18 bins from 81.25°S to 81.25°N with step 10°, and searched for the absolute minimum, absolute maximum, averaged over the given period (Ave), and the standard deviations (SD) for those bins. Therefore selection of such narrow latitude bands (±1°) is one of the differences among other similar studies.

The selected three latitudes (80°S, 0°, 80°N) represent extreme distinct geographic locations. This is the first study to compare temperature and water vapor variability for 22 years of the SABER instrument. We processed hundreds of monthly data sets for all three selected latitude bins (for temperature and WV). The majority of the past studies focused on one variable (temperature or water vapor) for a limited time or over a specific location. The inclusion of mid-latitude region may be the focus of my future work.

We have replaced the word "whole mesopause" with global mean in the revised text. We updated this information in the text, and figure captions. Here is an example (lines 115-116 of the revised manuscript)

**"All latitudes, and longitudes of the mesopause covered by the SABER instrument during a year are represented by global mesopause."**

We agree that creating a global mean temperature can be misleading since this approach combines different seasons in both hemispheres and SABER will include biases introduced by the changing latitude coverage with month (and year). And we are aware that the best "global" representation of SABER data would be 52°S – 52°N, as has been done by previous authors (e.g; Forbes et al., 2021; Liu et al., 2017; Mlynczak et al., 2022; Das et al., 2021). We have included an additional section of uncertainty (section 6) in this manuscript and provide relevant uncertainties of this work in that section. The relevant point of section 6 (lines 741-745 of the revised manuscript) is given below

**Creating a global mean temperature can be slightly misleading since this approach combines different seasons in both hemispheres (NH and SH), and for SABER it includes biases introduced by the changing latitude coverage with month (and year). For this reason, the best "global" representation of SABER data is 52°S – 52°N, as has been done by previous authors.**

Reference

➢ Das, U., 2021. Spatial variability in long-term temperature trends in the middle atmosphere from SABER/TIMED observations. Adv. Sp. Res. 68, 2890–2903. https://doi.org/https://doi.org/10.1016/j.asr.2021.05.014
➢ Forbes, J.M., Zhang, X., Randall, C.E., France, J., Harvey, V.L., Carstens, J., Bailey, S.M., 2021. Troposphere-mesosphere coupling by convectively forced gravity waves during Southern Hemisphere monsoon season as viewed by AIM/CIPS. J. Geophys. Res. Sp. Phys. 126, e2021JA029734. https://doi.org/https://doi.org/10.1029/2021JA029734
➢ Liu, X., Yue, J., Xu, J., Garcia, R.R., Russell III, J.M., Mlynczak, M., Wu, D.L., Nakamura, T.,

2017. Variations of global gravity waves derived from 14 years of SABER temperature observations. J. Geophys. Res. Atmos. 122, 6231–6249. https://doi.org/https://doi.org/10.1002/2017JD026604

➢ Mlynczak, M.G., Hunt, L.A., Garcia, R.R., Harvey, V.L., Marshall, B.T., Yue, J., Mertens, C.J., Russell III, J.M., 2022. Cooling and contraction of the mesosphere and lower thermosphere from 2002 to 2021. J. Geophys. Res. Atmos. 127, e2022JD036767. https://doi.org/https://doi.org/10.1029/2022JD036767

➢ Zhao, X. R., Sheng, Z., Shi, H. Q., Weng, L. B., and Liao, Q. X.: Long-term trends and solar responses of the mesopause temperatures observed by SABER during the 2002–2019 period, J. Geophys. Res. Atmos., 125, e2020JD032418, 2020.

**(R1-C10)** 5) Altitudes used in the study: It is not clear what altitudes were used for the T and H2O shown in the paper. I think it might be an average for 80-100 km (based on the legend in Fig 1), but it is not really stated clearly anywhere. This is an important point because the SABER errors increase rapidly with height. Furthermore, you should probably not mix measurements below and above the mesopause in a single average. In any case the paper should describe, and justify, the altitudes examined. Additionally, the Authors should consider looking at all altitudes.

Response to (R1-C10):

A constant altitude range of 80-100 km is used throughout the work. We have mentioned this information in multiple places in the revised manuscript and the same is given below

(lines 102-104 of the revised manuscript):

**"Discussion related to an analysis of 22 years of monthly temperature and WV profiles in the mesopause region (80-100 km altitude) are investigated."**

(lines 111-112 of the revised manuscript):

**"Spatial and temporal variations in long-term temperature and WV are analyzed in the mesopause region (80-100 km altitude)."**

(lines 250-252 of the revised manuscript):

**"These references focused on specific altitudes, and latitude ranges of the mesopause however, our mentioned results in this section focused on 80-100 km constant altitude of the mesopause."**

(lines 753-756 of the revised manuscript):

**"Our global mean temperature and WV content may mix measurements below and above the actual dynamic mesopause in a single average, because our measurements are based on a constant altitude range (80-100 km), throughout the study period."**

We have included text related to altitudes and uncertainty in the revised manuscript (lines 753-756 of the revised manuscript) and the same is given below.

**"Our global mean temperature and WV content may mix measurements below and above the actual dynamic mesopause in a single average, because our measurements are based on a constant altitude range (80-100 km), throughout the study period."**

Additionally revised figures 2, 10, and 11 show spatial and temporal variability of temperature and WV at different altitude ranges of the mesopause region. Figure 2 (page 11 of the revised manuscript) is given  below as an example

[Figure]

**Figure 2.** Temperature and water vapor gradient between 80-100 km altitudes from SABER observations at the three selected latitude bins during 200-2023. a) Equator (0° ± 1º). b) Northern hemisphere (80°N ± 1º). (c) Southern hemisphere (80°S ± 1º), in the indicated months, by averaging all January, June, and September values from 2002 to 2023.

5)Table 1: There appears to be a wealth of useful information here, but the results are poorly described and somewhat confusing. For example, the list appears to contain both absolute values of T and H2O, in addition to trends in these quantities. The trends are also listed alternately as per year or per decade, and this needs to be rectified. Results are also given for a wide range of altitudes, which is problematic given the strong altitude dependence in T and H2O in the mesopause region. In addition, numerous investigations have shown that trends in the upper mesosphere vary strongly with height (references listed in this paper), and can even change sign at roughly NLC altitudes (depending on latitude and season). Given these complexities, the presentation of results in Table 1 needs to be substantially revised, including a consideration of the altitude dependence. Perhaps these results would lend themselves to being plotted vs. height instead. Finally, Table 1 neglects the H2O trends derived from SABER and MLS observations by Yue et al. (2022, GRL; reference given in this paper), which are particularly relevant to the present study.

Response to (R1-C11):
We have updated Table 1 as suggested by the three reviewers (page number 14-16 of the revised manuscript) and the same is given below. Table 1 has more relevant details than the previous version of Tabl 1.

Table 1. Temperature and water vapor content comparisons with past studies in mesopause

| Trend K/decade | Avg. Temp | Altitude (km) | Location/Season/Data source | References |
|---|---|---|---|---|
| Temperature | | | | |
| Min: | 184.54 K | 80-100 | Global/summer (Jun. and Jul.)/ SABER | This study |
| 0 | 188.20 K | | Global/winter (Jan. and Dec.)/ SABER | |
| | 162.64 K | | 80°N ± 1º / summer (Jun. and Jul.)/SABER | |
| | 201.14 K | | 80°N ± 1º / winter (Jan. and Dec.)/SABER | |
| Max: | 193.21 K | | 80°S ± 1º / summer (Jun. and Jul.)/SABER | |
| -1.21 | 161.14 K | | 80°S ± 1º / winter (Jan. and Dec.)/SABER | |
| | 185.81 K | | 0° ± 1º / summer (Jun. and Jul.)/SABER | |
| | 188.95 K | | 0° ± 1º / winter (Jan. and Dec.)/SABER | |
| Min: 0 | 130-190K | 80-100 | 83°N to 83°S- all latitudes/ SABER | Zhao et al., 2020 |
| | 188±2 K | | 83°N/Northern hemisphere / SABER | |
| Max: | 135±2 K | | 83°S/Southern hemisphere / SABER | |
| -1.4 | 158±2 K | | 0°/Equator / SABER | |
| | 139 K | 90 | 80°S/January/ SABER | Wang et al., 2022 |
| | 180 K | 86 | 40°S/January/ SABER | |
| | 129 K | 90 | 80°N/July/ SABER | |
| | 161 K | 83 | 55°N/July/ SABER | |
| | 160 K | ~100 | 30°N/ around equinoxes (March)/ SABER | Xu et al., 2007 |
| | 185 K | ~80 | 30°N/ around equinoxes (March)/ SABER | |
| | 124 K | ~100 | 80°N /solstice period (June)/ SABER | |
| | 135 K | ~80 | 80°N /solstice period (June)/ SABER | |
| | 133 K | ~100 | 80°N /solstice period (December)/ SABER | |
| | 143 K | ~80 | 80°N /solstice period (December)/ SABER | |
| | ~126 K | 80-100 | Summer polar region/ SABER | |
| | ~190 K | 80-100 | Winter polar region/ SABER | |
| | 156-162 K | 84 | 45–50°N/ summer night time (Aura/MLS) | Dalin et al., 2023 |
| | 152-157 K | 84 | 50–55°N/ summer night time (Aura/MLS) | |
| | 147-151 K | 84 | 55–60°N/ summer night time (Aura/MLS) | |
| | 151-159 K | 89 | 45–50°N/ summer night time (Aura/MLS) | |
| | 147-153 K | 89 | 50–55°N/ summer night time (Aura/MLS) | |
| | 141-146 K | 89 | 55–60°N/ summer night time (Aura/MLS) | |

| | | | | |
|---|---|---|---|---|
| -2.5 | ~177.6 K | 97 | 41°N - 42°N / non summer months /Na lidar | Yuan 2019 |
| -2.3 | ~177.6 K | 92 | 41°N- 42°N /non winter months / Na lidar | |
| -3.8 | ~177.6 K | 97 | 41°N - 42°N / winter /Na lidar | |
| -1.75 | ~177.6 K | 92 | 41°N- 42°N /summer / Na lidar | |
| -2.3 | 160-230 K | 87 | 51°N/ all seasons/SABER instrument | Offermann et al., 2010 |
| Up | 158-238 K | 87 | 51°N/ all seasons/OH | |
| To | 160-232 K | 87 | 48°N/ all seasons/SABER instrument | |
| -6.0 | 145-235 K | 87 | 48°N/ all seasons/ OH | |
| -6.8 | | ~100 | 41°N (Lidar + SABER + Model) | She et al., 2009 |
| -1.5 | ~184K | ~91 | 41°N/January  (Lidar + SABER + Model) | |
| - 0.64 | ~200 K | ~85 | 41°N/January  Na lidar | |
| -0.64 | 160-245 K | 85-86 | 41°N/all seaons/Na lidar | She et al., 2015 |
| -2.8 | 160-235 K | 91-93 | 42°N/all seasons/Na lidar | |
| -0.23 | | 87 | 69°S/winter /Hydroxyl airglow | French et al., 2005 |
| -0.5 | | 80-95 | ±52° latitude  (WACCM-Model) | Garcia et al., 2019 |
| -2.4 | 160-173 K | 80-100 | ~57°N / summer mesopause (ground based) | Dalin et al., 2020 |
| -0.4 | 202-218 K | 80-100 | ~57°N/winter (ground based) | |
| -0.89 | 194-202 K | 87 | 51° N/annual mean | Kalicinsky et al., 2016 |
| | 185-201 K | 87 | 48∘ N/annual mean | |
| -4.0 | 135 K | 90 | 78⁰N/summer MLS on the Aura satellite. | Hall et al., 2012 |
| | ~200 K | 90 | 78⁰N/winter / radar observation | |
| -1.2 | 146-154 K | 83 | 55–61°N/ annual /LIMA and MIMAS model | Lübken et al., 2018 |
| -2.9 | 160-230 K | 98.5 | 41°N/all season/ Na lidar | She and Krueger, 2004 |
| -2.1 | 198-228 K | 80-100 | 63°N/ January/ SABER | Ammosov et al., 2014 |
| | 196-215 K | 80-100 | 63°N/ February/ SABER | |
| -2 | | 80-100 | middle & subpolar latitudes /summer/ model | Grygalashvyly et al., 2014 |
| -0.5 | | | middle & subpolar latitudes/ winter/ model | |
| -2.2 | | 80-100 | Middle latitudes/Airglow measurement | Perminov et al., 2014 |
| -0.24 | 140-170 K | 80 - 84 | 64–74°N/ all season/SOFIE | Hervig et al., 2015 |
| -0.5 | 145-166 K | 80-84 | 77°N /Satellite instrument and Model | Hervig et al., 2016 |
| -1.2 | 140-220 K | 87 | 68°S/ winter/ OH nightglow | French et al., 2020a |
| -0.3 | ~196 K | 87 | 23-26°S /March-April/SABER & airglow | Noll et al., 2017 |
| | 145-235 K | ~87 | 74°N /spectrometric observations of the OH | Medvedeva and Ratovsky, 2023 |

| Water vapor mixing ratio | | | | |
|---|---|---|---|---|
| ~1.30 ppmv | 80-100 | Global/summer/ SABER | This study |
| ~1.20 ppmv | | Global/winter/ SABER | |
| ~1.90 ppmv | | 80°N ± 1⁰ /summer/SABER | |
| ~0.49 ppmv | | 80°N ± 1⁰ /winter/SABER | |
| ~0.67 ppmv | | 80°S ± 1⁰ / summer/SABER | |
| ~2.30 ppmv | | 80°S ± 1⁰ /winter/SABER | |
| ~1.20 ppmv | | 0° ± 1⁰ /summer/SABER | |
| ~1.10 ppmv | | 0° ± 1⁰ /winter/SABER | |
| 4.2-5.1 ppmv | 84 | 45–50°N/ summer night time (Aura/MLS) | Dalin et al., 2023 |
| 4.5-5.4 ppmv | 84 | 50–55°N/ summer night time (Aura/MLS) | |
| 4.7-5.6 ppmv | 84 | 55–60°N/ summer night time (Aura/MLS) | |
| 3.1-3.6 ppmv | 89 | 45–50°N/ summer night time (Aura/MLS) | |
| 3.3-3.9 ppmv | 89 | 50–55°N/ summer night time (Aura/MLS) | |
| 3.3-3.9 ppmv | 89 | 55–60°N/ summer night time (Aura/MLS) | |
| 1-8 ppmv | 80 - 84 | 64–74°N/ all season/SOFIE | Hervig et al., 2015 |
| 5.4-5.8 ppmv | 80-84 | 77°N /Satellite instrument and Model | Hervig et al., 2016 |
| 0-7.0 ppmv | 80-100 | 66°-79°N/SOFIE on AIM & ALOMAR lidar | Hervig et al., 2009a |
| 1-2 ppmv | 95 | 66°-79°N/satellite measurement | |
| 1 ppmv | 90 | 78°N/ summer/1-D model | Murray and Jensen, 2010 |
| 3 ppm | 86 | 67.9°N- Polar region/Summer/Model | Gumbel et al., 2003 |
| 2.3 ppmv | 85 | Mid-latitude /Jul./Ground-based microwave | Bevilacqua et al., 1983 |
| 1.6 ppmv | 85 | Mid-latitude /Sep./Ground-based microwave | |
| 1.0 ppmv | 85 | Mid-latitude /Jan./Ground-based microwave | |
| 1.2 ppmv | 85 | Mid-latitude /Apr./Ground-based microwave | |
| 0.1 ppmv | 85 | Mid-latitude /Dec./Ground-based microwave | |
| 1.1 ppmv | 85 | Mid-latitude /Apr./Ground-based microwave | |

| | | | |
|---|---|---|---|
| 0-7.0 ppmv | 80-100 | 66°-79°N/SOFIE on AIM & ALOMAR lidar | Hervig et al., 2009a |
| 1-2 ppmv | 95 | 66°-79°N/satellite measurement | |
| 1.5-4.5 ppmv | 80 | 69ºN/Ground-based microwave | Seele and Hartogh, 1999 |
| 2-2.5 ppmv | 85 | Polar summer/Jun., Jul., Aug.,/ground-based | |
| 0.2 ppmv | 84 | 67ºN/3-D model / | Von Zahn & Berger, 2003 |
| 3 ppmv | 80-83 | 50ºN-80ºN/3-D model / | |
| ~2.0 ppmv | 80-83 | 50ºN-80ºN/3-D model / | |
| ~1.5 ppm | 90 | 72.5ºN /Jul., Aug./3D-Model and HALOE | Körner & Sonnemann, 2001 |
| ~3.5 ppm | 85 | 72.5ºN /Jul., Aug./3D-Model and HALOE | |
| ~5.1 ppm | 80 | 72.5ºN /Jul., Aug./3D-Model and HALOE | |
| 1.0 ppmv | ~83 | 65°–70°N /winter / HALOE measurement | Hervig et al., 2003 |
| 8.0 ppmv | ~83 | 65°–70°N /summer / HALOE measurement | |
| 0.45 - 4.81 ppmv | 80-94 | 78°N/ summer/Model, | Lubken et al., 2004 |
| ~4.5 ppmv | 80 | 78°N/ summer/Model, | |
| 3.4 ppmv | 85 | 78°N/ summer/Model, | |
| 1.98 ppmv | 90 | 78°N/ summer/Model, | |
| 2-4 ppmv | 82 | 55°N–55°S/SABER | Yue et al., 2019 |
| ~3.5 ppmv | 80 | 19.5°N/ Sep./Spectrometer mouna | Nedoluha et al., 2022 |

**(R1-C12a)** 6) Figure 7: These results are not described very well. What altitude is this for? Are you comparing Northern March to Southern March and Northern Sep. to Southern Sep.? If so then the differences are not meaningful as they are for two different seasons (e.g., spring vs. fall). Also, the diagram of Earth's orbital positions is not needed.

Response to (R1-C12a):
We used a constant altitude (80-100 km) throughout the study period as explained in the above response (R1-C10). We also include altitude-related information in the caption of Figure 7 (line numbers: 534-536).

**Figure 7.** Difference in NH-SH temperatures and NH-SH, WV at 80-100 km altitude during selected months of equinoxes (March and September) and solstices (January and July). a, c) Water vapor content difference. b, d) Temperature difference.

In Figure 7 we showed the annual difference in temperature between NH and SH during March. Similarly the annual difference in temperature between NH and SH during September. We have revised Figure 7 as suggested. We improved the text and described the information in clear statements as compared to the previous version of the manuscript (page 26 of the revised manuscript) and the same is given below.

Revised Figure 7:

[Figure]

**Figure 7.** Difference in NH-SH temperatures and NH-SH, WV at 80-100 km altitude during selected months of equinoxes (March and September) and solstices (January and July). a, c) Water vapor content difference. b, d) Temperature difference.

**(R1-C12b)** 6) Figure 7: Finally, a better illustration of these results would be a line plot of difference vs. year.

Response to (R1-C12a):
A line plot of differences vs. year is already provided in the next Figure (Figure 8)

**(R1-C13)** 7) Figure 8: Why are you comparing the temperature and H2O, for summer and winter, in different hemispheres? If your aim is to illustrate the seasonality then do it in the same hemisphere. If you are concerned with differences between hemispheres, then compare

the same seasons in the north and south (e.g., for summer use June in the north vs. December in the south). Later in the text (line 421-423) you quote large seasonal differences of 156, 210, and 186K. Nowhere in the results are such differences evident, and this should be checked.

Response to (R1-C13):

In Figure 8, we are presenting inter-annual variations in monthly mean temperature and water vapor from SABER observations over selected bins of latitudes during 2002-2023. For example, Figure 8a shows 22 years of July mean temperature for each individual year. Figure 8a also compares July's mean temperature in different Hemispheres as shown below.

[Figure]

(Figure 8a - July)

This is also a kind of compression during the same seasons in the north and south as you suggested [e.g., for summer use July in the north (Figure 8a, red line) vs. January in the south (Figure 8b, blue line)], as mentioned below

[Figure]

(Figure 8a - July)          (Figure 8b - January)

Readers can compare similar hemispheric comparison in the revised Figure 2, and we have included relevant text in the revised manuscript (lines 566-567) as given below

**"The vertical temperature and WV gradients during June at NH (Figure 2b) are quite similar to the vertical temperature and WV gradients during January at SH (Figure 2c)."**

[Figure]

Later in the text (line 421-423)

The quoted line (previously 421-423) has been replaced by a new sentence (lines 649-650 of the revised manuscript) and the same is given below.

**"SH is warmer than NH during July (~29 K) and September (~5 K) and colder than NH during January (~40.6 K) and March (~6.3 K)."**

**(R1-C14)** 8)Throughout the article you refer to the "north pole" and "south pole", yet your results are for 80°N and 80°S, which are not the poles. Please be specific and use the nomenclature 80°N and 80°S.

Response to (R1-C14):

Agree, we have used a specific nomenclature as suggested (lines 112-115 of the revised manuscript) as given below

**"Spatially the region is divided into three latitude bins (Equator, Northern, and Southern Hemispheres). Taking temperature and WV data in 0° ± 1º, 80°N ± 1º, and**

**80°S ± 1° latitude bins represent the equator, northern hemisphere (NH), and southern hemisphere (SH) respectively."**

And then we used this nomenclature throughout the text of the revised manuscript.

**(R1-C15)** 9)Figure 10: The results are very hard to interpret, please try another approach.

Response to (R1-C15):

Agree, we have removed longitude in the revised Figures. We introduced two figures instead of one figure which displays relatively better information related to spatial variability of temperature and WV(page number 34 of the revised manuscript) and the same is given below

[Figure]

**Figure 10**.  Two-dimensional (latitude, and altitude) variation in temperature (K), and WV (ppmv) at three latitudes as indicated, and three altitudes (80 km, 90 km, and 100 km)  during January 2003.

Thanks to anonymous reviewer 1 for his/her constructive comments and suggestions.

------------------------ End of the response to reviewer 1 ------------------------

---

## Author Comment (AC2)

**Response to Reviewer 2 comments**
**Manuscript Number: EGUsphere-2024-1144**
**Manuscript title: Spatial and temporal variation in long-term temperature and water vapor in the mesopause Region, by Chaman Gul et al.,**

30$^{th}$ July 2024

Dear anonymous reviewer,

Thanks for the comments, suggestions, and recommendations for the EGUspher-2024-1144 manuscript. Comments are constructive and we quite improved the manuscript after addressing all the comments. We have thoroughly considered and carefully addressed all issues mentioned in the comments and have properly outlined every single change made in response to reviewer comments as suggested. We have made the required corrections in the revised manuscript (visible in tracked change mode) and prepared a list of point-by-point responses as given below starting from page #2 of this document. We have attached two copies of the revised manuscript, one with track change mode having all edits/corrections and the other is a fair copy of the manuscript where we have accepted all the mentioned edits/corrections. The reviewer's comments are in **black** text, the author's responses are in **blue** text, the modified/corrected text from the revised manuscript is in bold **brown** text, and references are in green text. Modified line numbers are in **yellow highlighted** text.

**Response to reviewer 2 (R2) comments (Cs):**

**(R2-C1)**This paper has interesting topic, which is well in scope of ACP. However, there are various flaws in the paper. English of the paper needs substantial improvement; some suggestions are below. I recommend major revision.

Response to (R2-C1):

Thank you very much for your precious time and constructive comments. We have modified /revised the manuscript (including the language) based on the reviewers' comments. We have made the required corrections in the revised manuscript (visible in tracked change mode) and prepared a list of point-by-point responses as given below.

**Comments:**

**(R2-C2)** The results of this paper generally confirm previous findings with longer datasets analyzed here. Authors should clearly describe in Conclusions, what is new in their results compared to the current state-of-the-art.

Response to (R2-C2): Numerous studies have examined the SABER data set previously to investigate temperature or WV. This article investigates long-term changes in temperature and WV (both) and their long-term comparison within a unique selection of time and space domains. We think the selected narrow latitude bins from each selected geographical location, excluding transitional months, and inclusion of high latitude regions (beyond ~53ºN or ~53ºS) from both hemispheres make this article different from other previous works. The majority of the past studies focused on one variable (temperature or water vapor) for a limited time or over a specific location. This is the first study to compare temperature and water vapor variability for 22 years of the SABER instrument. We processed hundreds of monthly data sets for all three selected latitude bins (for temperature and WV). Multiple studies (e.g; Forbes et al., 2021; Liu et al., 2017; Mlynczak et al., 2022; Das et al., 2021) are limited to latitude band ~50°S to ~50°N, mainly due to TIMED ~60 days yaw cycle. In the present study, we have included high-latitude regions from both hemispheres along with some missing data. Our results generally showed similar seasonality and trends (as presented in the past), but different in magnitude. We have described these similarities and dissimilarities in multiple places of the revised manuscript including the conclusion sections (lines 795-802 of the revised manuscript) and the same is given below

"**The selected narrow latitude bins (2º each) at the three extreme geographical positions (NH, SH, equator), excluding transitional months, use of monthly SABER data set, and use of constant altitude range (80-100 km) throughout the study period made our results slightly different in magnitude as compared to the past reported results. Very few researchers (e.g: Hervig et al., 2015) focused on both temperature and water vapor. Therefore, our temperature and water vapor results, obtained from 22-year SABER**

observations, are expected to be a robust measure of the mesopause temperature and water vapor variability.”

**(R2-C3)** You are working with monthly data. However, trends based on SABER monthly data are not correct, trends should be based on data averaged over yaw cycle of SABER/TIMED.

Response to (R2-C3): In the revised text trends for three selected latitude bins (~0° ± 1°, ~80° ± 1°N, and ~80° ± 1°S ) are based on the data averaged over the yaw cycle of SABER/TIMED (Figure 1) along with some limitation discussed in section 6 of the revised manuscript. We agree that using monthly data for high-latitude regions is a source of uncertainty in results and the best "global" representation of SABER data is 52°S – 52°N, as has been done by previous authors. We have included sections explaining the yaw cycle, and availability of data at high latitudes, particularly related to this paper dataset. Additionally, we have included a section on associated uncertainty and limitations in this manuscript and provide relevant uncertainties in that section.

Section 2.2 of the revised manuscript (pages 26 and onward, of the revised manuscript) describes the yaw cycle of the SABER instrument as given below.

**"2.2. TIMED-SABER instrument**

**The TIMED-SABER satellite views 90° to the right of the velocity vector of the TIMED spacecraft, and completes a full 24-hour local time coverage in 60-63 days (Russell III et al., 1999; Mlynczak et al., 2003; Figure 1). The SABER instrument scans the atmosphere from the troposphere up to the lower thermosphere and obtains vertical profiles kinetic temperature and volume mixing ratio of WV (Russell et al., 1999). The instrument performs near-global measurements and provides an excellent quality of the measured infrared limb radiances (Esplin et al., 2023). Technical description of the SABER instrument and further relevant information are discussed by Mlynczak, (1997) and Russell III et al. (1999). TIMED satellite rotates 180° about its yaw axis and provides latitude coverage continuously in the range of 53°S to 83°N and then switching to 83°S to 53°N every ~60 days (Russell III et al., 1999). Due to the asymmetrical latitudinal coverage of the SABER instrument, there are some missing measurement months at high latitudes (52°N-83°N or 52°S-83°S). Multiple studies ( e.g; Forbes et al., 2021; Liu et al., 2017; Das, 2021) are limited to the latitude band ~50°S to ~50°N, mainly due to the TIMED ~60 days yaw cycle. In the present study, we have included high-latitude regions from both hemispheres along with some missing data. For example, coverage of high northern latitudes included July in the early years, but not**

**during the recent several years (2017-2023)………"** Please have a look at this section in the revised manuscript for full details.

Revised Figure 1: (page # 7 of the revised manuscript)

[Figure]

**Revised Figure 1**. SABER instrument latitude coverage versus time for observation. a) Monthly data coverage in selected months versus latitude ranges from January 2002 to December 2023, excluding transitional months. b) Comparison of SABER latitude coverage and monthly data versus time during years (2002-2003). c) Typical temporal coverage of TIMED-SABER instrument measurements. d) Latitude versus longitude tangent point locations for one day of observations in its north viewing phase (83°N to 52°S) – a north viewing yaw mode.

Uncertainties related to high latitude regions: (line numbers 713 and onward )

**"Section 6. Associated uncertainties and limitations**

The possible sources of uncertainties during the analysis of long-term temperature and WV are mentioned below.

1. Large uncertainty is related to the analysis of temperature and WV over SH and NH (above ~53⁰ latitudes) and has a relatively larger bias in results as compared to the results over the equator. The yaw cycle is ~60 days, and only one polar region (SH or NH) is observed in each yaw cycle, and the selected polar regions are only alternatively observed half of a year owing to the yawing of the TIMED satellite. In other words, the latitudinal coverage is governed by a 60-day yaw cycle that allows observations of latitudes from 83⁰S to 52⁰N in the south-viewing phase or from 53⁰S to 82⁰N in the North-viewing phase (further details are given in the text). Multiple studies (e.g; Forbes et al., 2021; Liu et al., 2017; Mlynczak et al., 2022; Das, 2021) are limited to the latitude band ~50°S to ~50°N. In the present study, we have included high-latitude regions from both hemispheres along with some missing months. Missing months are usually April, August, or December in the NH and February, June, or October in the SH. As a result, the choice of these months for high latitudes introduces a systematic bias in the time series.

2. Temperature and WV trends over NH and SH are calculated for six months because April and December data were insufficient for long-term trends over NH. Similarly, June and October data was limited for SH trend estimation. Therefore, trends over the equator are more accurate than those of NH and SH trends.
   "

So, we present our results along with the above-mentioned uncertainties in the revised text.

**(R2-C4)**Page 6: Shorter-term decreases and increases of temperature reflect primarily the 11-year solar cycle.

Response to (R2-C4): Agree we have included the recommended sentence as suggested (lines 239-241 of the revised manuscript) and the same is given below

"A second decrease in temperature by ~4 K was observed from 2014 to 2018. A decrease of ~0.37 K and ~0.14 K was observed during 2002-2018, and 2002-2023 respectively. The cyclic temperature variations reflect primarily the 11-year solar cycle."

**(R2-C5)** Lines 166-167: The greater solar flux in December/January due to orbital eccentricity contributes to difference in temperature for sure.

Response to (R2-C5): We have modified the sentences as suggested (lines 281-282 of the revised manuscript) and the same is given below

**"The greater solar flux in December/January than in June/July is due to the Earth's orbital eccentricity, as discussed by Chu et al. (2003)."**

**(R2-C6)** Lines 195-197: Variations of temperature with height are similar in June and December but evidently different in April (Fig. 2c). Correct your sentence.

Response to (R2-C6): We have removed these lines from the revised text, and restated these sentences according to the revised Figure 2.

**The vertical profiles of annual mean temperature and WV gradient (vertical profiles with respect to changing altitude) are plotted as a function of year in Figure 2. Plots in Figure 2 are for three latitude bins NH, SH, and equator during three months (January, June, and September). We obtained mean temperature and mean WV content for these months by averaging all January, June, and September values from 2002 to 2023. We did a similar 22-year average for other months (March, April, July, and October) but not shown in Figure 2. There is an inverse relation between temperature and WV. An anticorrelation between WV with the solar cycle was also shown by Yue et al. (2019); Dalin et al. (2023) and Hervig and Siskind, (2006). The precise relationship between WV saturation mixing ratios and cold point temperature depends upon the temperature as well as exact pressure (altitude), with Seidel et al. (2001) giving a value of ~0.6 ppmv/K, Fueglistaler and Haynes, (2005) ~0.5 ppmv/K, and Nedoluha et al. (1998) ~0.7 ppmv/K. In the present study, the maximum and minimum WV change between 81-100 km altitude was ~4.3 and ~1.6 ppmv respectively.**

**(R2-C7)** Table 1: Table 1 is amazing collection of trend information. Differences in temperature trends may be partly from different changes of ozone in different periods. Water vapor – different periods may include or not water vapor drops in 2001-2002 and 2014, which affects trends.

Response to (R2-C7): Agree we updated the sentence as suggested (Line number 357-360 of the revised manuscript) and the same is given below

**"References shown in Table 1 are focused on temperature and wv variation at different latitudes and altitude ranges of the mesopause region. Differences in temperature trends may be partly from different changes of ozone in different periods. Different periods**

**may include or not WV drops in 2001-2002 and 2014 (solar maxima's), which affects WV trends given in Table 1."**

**(R2-C8)**Lines 298-300: I do not understand these two sentences. What would you like to say?

Response to (R2-C8): Sorry for writing a confused sentence. We have replaced the sentence with a new sentence, in a new position (lines 387-390 of the revised manuscript) and the same is given below

**"Generally, temperature decreases with increasing altitude however, this temperature gradient is small during June and July as compared to other selected months. Temperature decreased from 80 to 100 km altitude by 10 to 20 K during January, June, and September (Figure 2a). "**

**(R2-C9)** Lines 83-84: WV controls the concentration of $O_3$ – add at least one reference.

Response to (R2-C9): Agree, we included a reference as suggested (lines 81-82 of the revised manuscript) and the same is given below

**"WV content in the atmosphere controls the concentration of ozone that, in turn, affects mesospheric cooling (Smith, 2004)."**
Reference

Smith, A. K.: Physics and chemistry of the mesopause region, J. Atmos. solar-terrestrial Phys., 66, 839–857, https://doi.org/https://doi.org/10.1016/j.jastp.2004.01.032, 2004.

**(R2-C10)**Line 330 and 331: ""June showed" should be "June and July showed"; "seven months" should be "six months (Figure 5)".

Response to (R2-C10): Agree, sentence modified as suggested (lines 437-438 of the revised manuscript) and the same is given below

**"June and July temperature gradients are different than the vertical temperature gradients of other selected six months (Figure 2b)."**

**(R2-C11)**Lines 335-338: Where these statements are documented/illustrated in the paper?

Response to (R2-C11): We couldn't show vertical profiles of all selected months in previous Figure 2. Therefore we analyzed for indicated months (lines 335-338) but not shown in

Figure 2. In the revised Figure 2 we have increased the number of months (January, June, and September). But again information related to March is given in the text but not shown in Figure 2, so we slightly changed the sentence (lines 443-444 of the revised manuscript)as given below

**"There is a clear temperature decrease between 84 km (~202 K) to 96 km (~172 K) during September (Figure 2). The mean temperature at 80 km during March was ~210 K and decreased to ~185 K at an altitude of 100 km (March is not shown in Figure 2)."**

Revised Figure 2 is given below:

[Figure]

**Figure 2.** Temperature and water vapor gradient between 80-100 km altitudes from SABER observations at the three selected latitude bins during 200-2023. a) Equator (0° ± 1º). b) Northern hemisphere (80°N ± 1º). (c) Southern hemisphere (80°S ± 1º), in the indicated months, by averaging all January, June, and September values from 2002 to 2023.

**(R2-C12)** Lines 352-353: This statement requires a citation.

Response to (R2-C12): We include a reference as suggested (lines 463-464 of the revised manuscript) and the same is given below

**"December and January; this is because, at middle and high latitudes, the general transport of H2O is directed upward in summer and downward in winter (Sonnemann et al., 2005)."**

Reference:

Sonnemann, G. R., M. Grygalashvyly, and U. Berger (2005), Autocatalytic water vapor production as a source of large mixing ratios within the middle to upper mesosphere, J. Geophys. Res., 110, D15303, doi:10.1029/2004JD005593.

**(R2-C13)** Lines 421-423: Differences 156, 210 and 183 k are nonsense and do not correspond to Fig. 8.

Response to (R2-C13): Agree, we have removed the text from the revised manuscript (lines 557-559 of the revised manuscript), and the removed sentence is visible in the track change version of the manuscript only.

**(R2-C14)** List of references:

Response to (R2-C14): We have rechecked and updated all references using Mendeley software and made all references according to the requirements of ACP.

- **(R2-C15)** Important reference Guo (2024) is missing.

Response to (R2-C15): Thanks for mentioning an interesting review paper "A review of atmospheric water vapor lidar calibration methods by "Guo et al., 2024". We cited this paper in the introduction part of the revised manuscript (lines 99-100 of the revised manuscript) and the same is given below

**"The latest progress and applications of atmospheric WV lidar calibration have been recently reviewed by Guo et al. (2024)."**

Reference

The latest progress and applications of atmospheric WV lidar calibration have been recently reviewed by Guo et al. (2024).

- **(R2-C16)** Wherever possible add either doi index or https address.

Response to (R2-C16): We have included the doi index or https address as suggested. A few examples are mentioned below

Proceedings of the NATO Advanced Study Institute held at Spåtind, Norway, April 12–22, 1977, 93–127, 1977. 10.1007/978-94-010-1262-1_10

Berger, U. and Lübken, F.: Mesospheric temperature trends at mid-latitudes in summer, Geophys. Res. Lett., 38, 2011. https://doi.org/10.1029/2011GL049528

Berger, U. and Von Zahn, U.: Icy particles in the summer mesopause region: Three-dimensional modeling of their environment and two-dimensional modeling of their transport, J. Geophys. Res. Sp. Phys., 107, SIA-10, 2002. https://doi.org/10.1029/2001JA000316

Bevilacqua, R. M., Olivero, J. J., Schwartz, P. R., Gibbins, C. J., Bologna, J. M., and Thacker, D. J.: An observational study of water vapor in the mid-latitude mesosphere using ground-based microwave techniques, J. Geophys. Res. Ocean., 88, 8523–8534, 1983. https://doi.org/10.1029/JC088iC13p08523

Bittner, M., Offermann, D., and Graef, H. H.: Mesopause temperature variability above a midlatitude station in Europe, J. Geophys. Res. Atmos., 105, 2045–2058, 2000. https://doi.org/10.1029/1999JD900307

Brasseur, G. and Solomon, S.: Aeronomy of the Middle Atmosphere, 452 pp., D, 1986. https://doi.org/10.1029/EO067i009p00114-03

Brasseur, G. P. and Solomon, S.: Aeronomy of the middle atmosphere: Chemistry and physics of the stratosphere and mesosphere, Springer Science & Business Media, 2005. 10.1007/1-4020-3824-0

Chabrillat, S., Kockarts, G., Fonteyn, D., and Brasseur, G.: Impact of molecular diffusion on the CO2 distribution and the temperature in the mesosphere, Geophys. Res. Lett., 29, 11–19, 2002. https://doi.org/10.1029/2002GL015309

Chandra, S., Jackman, C. H., Fleming, E. L., and Russell III, J. M.: The seasonal and long term changes in mesospheric water vapor, Geophys. Res. Lett., 24, 639–642, 1997. https://doi.org/10.1029/97GL00546

Wording or misprints:

- **(R2-C17)** Line 72: Start with "Water vapor (WV)"

Response to (R2-C17): Agree, sentences modified as suggested (lines 70 of the revised manuscript) and the same is given below

**"Water vapor (WV) is one of the strongest greenhouse gases and plays a crucial radiative balance role in the atmosphere. WV in the upper atmosphere can affect global surface climate (Solomon et al., 2010)."**

- **(R2-C18)** Line 84: "water-mixing" should be "water vapor-mixing"

Response to (R2-C18): Sentence removed on the recommendation of the reviewer 3 (R3-C2).

- **(R2-C19)** Line 91: reference should be "Chandra et al. (1997)"

Response to (R2-C19): The mentioned sentence is deleted from the revised text because the sentence was not fit in the flow of information.

- **(R2-C20)** Line 135: "(Figure 2)" should be "Figure 2"

Response to (R2-C20): Sentence removed from the revised manuscript.

- **(R2-C21)** Lines 174-176: :warming, and the causes adiabatic cooling" should be "warming and adiabatic cooling, respectively"

Response to (R2-C21): We have modified the sentences as suggested (lines 279-280 of the revised manuscript) and the same is given below

**"Downwelling in the winter hemisphere and upwelling in the summer hemisphere cause adiabatic warming, and adiabatic cooling, respectively"**

- **(R2-C22)** Line 236: "(Hervig et al., 2003) should be "Hervig et al. (2003)

Response to (R2-C22): Agree, citation corrected as suggested (lines 337-338 of the revised manuscript) and the same is given below

**"There are few studies including Hervig et al. (2003) which showed WV enhancement above 86 km altitudes."**

- **(R2-C23)** Line 246: "bysolar" should be "by solar"

Response to (R2-C23): Agree, the word changed as suggested (lines 343-344 of the revised manuscript) and the same is given below

**"At mesospheric heights, WV is strongly photo-dissociated by solar Lyman alpha (Brasseur and Solomon, 1986)."**

- **(R2-C24)** Line 248: "The solar" should be "the solar"

Response to (R2-C24): Sentence removed from the revised text.

- **(R2-C25)** Lines 264 and 267: "Figure 3" should be "(Figure 3)"; similarly Figure 4 on line 313, Figure 5c on line 360.

Response to (R2-C25): Section removed from the revised text. We rechecked the whole manuscript for similar mistakes and corrent it accordingly.

- **(R2-C26)** Line 265: "(Dalin et al., 2023)" should be "Dalin et al. (2023)"; similarly citations at line 278.

Response to (R2-C26): Section removed from the revised text. We rechecked the whole manuscript for similar mistakes and corrent it accordingly.

- **(R2-C27)** Line 323: "temperatures at" should be "temperatures (Figure 5) at"

Response to (R2-C27): Agree, sentence modified as suggested (lines 426-427 of the revised manuscript) and the same is given below

**"Similarly, monthly mean temperatures (Figure 5) at solstices (Jun/July and Dec/Jan) were ~162.64 K and ~201.14 K respectively,"**

- **(R2-C28)** Lines 325-326:"those Xu et al., 2007), showed" should be "those of Xu et al. (2007) showed"

Response to (R2-C28): We have modified the sentences as suggested (line 431 of the revised manuscript) and the same is given below

**"Our results are similar to those of Xu et al. (2007) showed a warmer mesopause at high latitudes during the December solstice than it is in the June solstice."**

- **(R2-C29)** Line 334: "altitude" should be "with altitude"

Response to (R2-C29): Agree, we have included "with" in the sentences as suggested (line 442 of the revised manuscript) and the same is given below

**"Average temperature during January was ~208 ± 5 K almost constant with increasing altitude and showed very little decrease in temperature (~8 K/20 km) with altitude in temperature."**

- **(R2-C30)** Line 350: "temperature" should be "WV"

Response to (R2-C30): Sorry for this mistake, we have corrected the sentences as suggested (lines 460-461 of the revised manuscript) and the same is given below

**"Monthly mean WV at solstices (Jun/July and Dec/Jan) was ~1.90 ppmv and ~0.49 ppmv respectively, indicating relatively high WV content during June and July and low during December and January"**

- **(R2-C31)** Line 352: "January this" should be "January; this"

  Response to (R2-C31): Sorry for this common mistake. Corrected as suggested (line 463 of the revised manuscript) and the same is given below

  **"indicating relatively high WV content during June and July and low during December and January; this is because, at middle and high latitudes, the general transport of H2O is directed upward in summer and downward in winter"**

- **(R2-C32)** Line 358: "WV 5a (yearly averaged)" should be "WV (Figure 5a, yearly averaged).

  Response to (R2-C32): We revised the text and removed the mentioned sentence, from the revised manuscript, visible in the track changed version.

- **(R2-C33)** Line 366: delete "of the"

  Response to (R2-C33): We have deleted "of the" from the sentences as suggested (lines 480 of the revised manuscript) and the same is given below

  **"There is a cooling trend in temperature (~0.58 K/decade) in the SH mesopause region."**

- **(R2-C34)** Line 368: "months and (December and January) were" should be "months, and December and January were"

  Response to (R2-C34): Agree, we modified the sentences as suggested (lines 480-481 of the revised manuscript) and the same is given below

  **"On average April (~197.64 K) followed by June and July were the hottest months, and December and January were the coldest months throughout the 22-year study period."**

- **(R2-C35)** Line 375: delete "was"

  Response to (R2-C35): Agree, we have deleted "was" as suggested.

- **(R2-C36)** Line 386: "ppmv respectively" should be "ppmv, respectively"

  Response to (R2-C36): We have modified the sentences as suggested (line 505 of the revised manuscript) and the same is given below

"The monthly mean WV at two equinoxes (Mar/Apr and Sep/Oct) was ~0.82 and ~0.54 ppmv lower than the monthly mean WV at solstices (Jun/July and Dec/Jan) which were ~0.67 ppmv and ~2.3 ppmv, respectively"

- **(R2-C37)** Line 388: "had relatively" should be "which however had relatively"

Response to (R2-C37): We have modified the sentences as suggested (lines 505-507 of the revised manuscript) and the same is given below

"This indicates relatively high WV content during summer (December and January). In the SH temperature is colder at mid-to-high latitudes during January (Wang et al., 2022), which however had relatively high WV content."

- **(R2-C38)** Line 399: "soloists" should be "solstices"

Response to (R2-C38): Agree, we have corrected the spelling as suggested (line 504 of the revised manuscript) and the same is given below

"The monthly mean WV at two equinoxes (Mar/Apr and Sep/Oct) was ~0.82 and ~0.54 ppmv lower than the monthly mean WV at solstices (Jun/July and Dec/Jan)"

- **(R2-C39)** Line 405: "however it look" should be "however the difference look"

Response to (R2-C39): We have modified the sentences as suggested (Line 540 of the revised manuscript) and the same is given below

"The difference between NH and SH temperature and WV at the solstice position is higher than the difference at the equinoxes, indicating that the magnitude of temperature and WV content near poles are relatively close at equinox positions however the difference looks to increase in the future (Figure 7)."

- **(R2-C40)** Line 414: "Intra-annual" should be "inter-annual"

Response to (R2-C40): We have revised the caption along with recommended changes as suggested (lines 550-551 of the revised manuscript) and the same is given below

"Figure 8. Inter-annual variations in monthly mean temperature and water vapor from SABER observations over selected bins of latitudes during 2002-2023."

- **(R2-C41)** Lines 429, 432-433: "(Wang et al., 2022)" should be "Wang et al. (2022); the same with Xu et al.

Response to (R2-C41): We have modified the references as suggested (lines 651-653 of the revised manuscript) and the same is given below

"**Wang et al. (2022) and Xu et al. (2007)** found that the mesopause during the June solstice is ~6–9 K colder than that during the December solstice. Huaman and Balsley, 1999 showed a predominant warmer SH"

We made similar changes throughout the manuscript, as suggested.

- **(R2-C42)** Line 435: "month North" should be "month for the North"

Response to (R2-C42): Agree, we have modified the sentences as suggested (lines 561-562 of the revised manuscript) and the same is given below

"**The winter solstice (January) was the higher temperature month for the NH and the lower temperature month for the SH (Figure 8).**"

- **(R2-C43)** Line 477: "(Dalin et al., 2020) should be "Dalin et al. (2020)"

Response to (R2-C43): We have updated the text as suggested (line 622 of the revised manuscript) and the same is given below

"**At the same time, Dalin et al. (2020) showed relatively stronger cooling at the summer mesopause (−2.4 K/decade).**"

- **(R2-C44)** Line 488: "in (Xu et al., 2007)" should be "by Xu et al. (2007)"

Response to (R2-C44): We have updated citations as suggested (lines 638-639 of the revised manuscript) and the same is given below

**A clear hemispheric asymmetry in temperature (Figure 7) was observed, possibly related to solar and gravity waves further discussed in Xu et al. (2007).**

- **(R2-C45)** Line 497: "; Xu" should be "and Xu". Similar at line 509

Response to (R2-C45): We have modified the text as suggested (lines 651-652 of the revised manuscript) and the same is given below

"**Wang et al. (2022) and Xu et al. (2007) found that the mesopause during the June solstice is ~6–9 K colder than that during the December solstice.**"

We applied similar changes in other places of the manuscript shown in the track change version of the manuscript.

- **(R2-C46)** Line 510: delete "almost"

Response to (R2-C46):

We deleted the whole sentence from the revised text because the sentence was presenting repeating information that was already stated.

- **(R2-C47)** Line 574: delete "year"

Response to (R2-C47): Agree, we have deleted "year" in the revised text as suggested (lines 786-789 of the revised manuscript) and the same is given below

**"Based on the monthly mean WV content for the selected eight months of analysis shows that 2018 had a relatively higher amount of WV content (~1.14 ppmv) followed by 2008 (~1.14 ppmv), and 2002 had the least amount of WV (~0.89 ppmv) followed by 2014 and 2003 (~1.0 ppmv)."**

Deletion is visible in the track change version of the manuscript.

Thanks to anonymous reviewer 2 for his/her constructive comments and suggestions.
----------------------- End of the response to reviewer 2 -----------------------

---

## Author Comment (AC3)

**Response to Reviewer 3 comments**
**Manuscript Number: EGUsphere-2024-1144**
**Manuscript title: Spatial and temporal variation in long-term temperature and water vapor in the mesopause Region, by Chaman Gul et al.,**

30[th] July 2024

Dear anonymous reviewer,

Thanks for the comments, suggestions, and recommendations for the EGUspher-2024-1144 manuscript. Comments are constructive and we quite improved the manuscript after addressing all the comments. We have thoroughly considered and carefully addressed all issues mentioned in the comments and have properly outlined every single change made in response to reviewer comments as suggested. We have made the required corrections in the revised manuscript (visible in tracked change mode) and prepared a list of point-by-point responses as given below starting from page #2 of this document. We have attached two copies of the revised manuscript, one with track change mode having all edits/corrections and the other is a fair copy of the manuscript where we have accepted all the mentioned edits/corrections. The reviewer's comments are in **black** text, the author's responses are in **blue** text, the modified/corrected text from the revised manuscript is in bold **brown** text, and references are in green text. Modified line numbers are in **yellow highlighted** text.

**Review of "Spatial and temporal variation in long-term temperature and water vapor in the mesopause Region" (egusphere-2024-1144) by Gul et al.**

**General Comments**

**(R3-C1)** This paper presents an analysis of SABER temperature and water vapor profiles in the mesopause region (80-100 km). Selected months (solstices, equinoxes) and geographic regions (Equator, North and South polar) are examined throughout the 22-year data record covering 2002-2023. Extreme values are determined on annual and monthly basis for geographic and seasonal comparisons.

This paper provides relatively little new information. Numerous studies have examined the SABER data set previously. It is not clear that the authors have adequately addressed the changes in SABER sampling over its long data record. The trend analysis is too simplistic, and does not consider multiple periodic forcing functions that also affect the long-term variations of temperature and water vapor.

Definite need for grammar to be polished. Not trying to address all such items in this review.

Response to (R3-C1):

Thank you very much for your precious time and constructive comments. This article investigates long-term changes in temperature and WV and its long-term comparison within a unique selection of time and space domains. We think selected narrow latitude bins from each selected geographical location, excluding transitional months, and inclusion of high latitude regions (beyond ~53ºN or ~53ºS) from both hemispheres make this article different from other previous works. The majority of previously published articles focused on temperature or water vapor. Multiple studies (e.g; Forbes et al., 2021; Liu et al., 2017; Mlynczak et al., 2022; Das et al., 2021) are limited to latitude band ~50°S to ~50°N, mainly due to TIMED ~60 days yaw cycle. In the present study, we have included high-latitude regions from both hemispheres along with some missing data.

 Very few researchers (e.g: Hervig et al., 2015) focused on both temperature and water vapor for relatively short periods. Therefore, our temperature and water vapor results, obtained from 22-year SABER observations, are expected to be a robust measure of the mesopause temperature and water vapor variability.

We have added information related to SABER instrument latitude coverage vs time (lines 126-175 of the revised manuscript) and the same is given below

**"2.2. TIMED-SABER instrument**

**The TIMED-SABER satellite views 90° to the right of the velocity vector of the TIMED spacecraft, and completes a full 24-hour local time coverage in 60-63 days  (Russell III et al., 1999; Mlynczak et al., 2003; Figure 1). The SABER instrument scans the**

atmosphere from the troposphere up to the lower thermosphere and obtains vertical profiles kinetic temperature and volume mixing ratio of WV (Russell et al., 1999). The instrument performs near-global measurements and provides an excellent quality of the measured infrared limb radiances (Esplin et al., 2023). Technical description of the SABER instrument and further relevant information are discussed by Mlynczak, (1997) and Russell III et al. (1999). TIMED satellite rotates 180° about its yaw axis and provides latitude coverage continuously in the range of 53°S to 83°N and then switching to 83°S to 53°N every ~60 days (Russell III et al., 1999). Due to the asymmetrical latitudinal coverage of the SABER instrument, there are some missing measurement months at high latitudes (52°N-83°N or 52°S-83°S). Multiple studies ( e.g; Forbes et al., 2021; Liu et al., 2017; Das, 2021) are limited to the latitude band ~50°S to ~50°N, mainly due to the TIMED ~60 days yaw cycle. In the present study, we have included high-latitude regions from both hemispheres along with some missing data. For example, coverage of high northern latitudes included July in the early years, but not during the recent several years (2017-2023)………" Please have a look at this section in the revised manuscript for full details.

Revised Figure 1: (page #7 )

[Figure]

**Revised Figure 1**. SABER instrument latitude coverage versus time for observation. a) Monthly data coverage in selected months versus latitude ranges from January 2002 to December 2023, excluding transitional months. b) Comparison of SABER latitude coverage and monthly data versus time during years (2002-2003). c) Typical temporal coverage of TIMED-SABER instrument measurements. d) Latitude versus longitude tangent point locations for one day of observations in its north viewing phase (83°N to 52°S) – a north viewing yaw mode.

Uncertainties related to high latitude regions: (line numbers 713 and onward )

**"Section 6. Associated uncertainties and limitations**

**The possible sources of uncertainties during the analysis of long-term temperature and WV are mentioned below.**

1. **Large uncertainty is related to the analysis of temperature and WV over SH and NH (above ~53⁰ latitudes) and has a relatively larger bias in results as compared to the results over the equator. The yaw cycle is ~60 days, and only one polar region (SH or NH) is observed in each yaw cycle, and the selected polar regions are only alternatively observed half of a year owing to the yawing of the TIMED satellite. In other words, the latitudinal coverage is governed by a 60-day yaw cycle that allows observations of latitudes from 83⁰S to 52⁰N in the south-viewing phase or from 53⁰S to 82⁰N in the North-viewing phase (further details are given in the text). Multiple studies (e.g; Forbes et al., 2021; Liu et al., 2017; Mlynczak et al., 2022; Das, 2021) are limited to the latitude band ~50°S to ~50°N. In the present study, we have included high-latitude regions from both hemispheres along with some missing months. Missing months are usually April, August, or December in the NH and February, June, or October in the SH. As a result, the choice of these months for high latitudes introduces a systematic bias in the time series.**

2. **Temperature and WV trends over NH and SH are calculated for six months because April and December data were insufficient for long-term trends over NH. Similarly, June and October data was limited for SH trend estimation. Therefore, trends over the equator are more accurate than those of NH and SH trends.**
**"**

So, we present our results along with the above-mentioned uncertainties in the revised text.

We used multiple linear regression analysis to determine temperature and water vapor trends from the observations. We derived trends using multiple linear regression with the inclusion of two terms, 1) the solar cycle (e.g., using Lyman – alpha) and 2) time (i.e., the trend). We have modified the sentences as suggested (lines 198-217 of the revised manuscript) and the same is given below

**"2.3.1. Multiple linear regression analysis**

**To investigate the long-term trends (temperature and WV) and the solar response of the mesopause temperature, a three-component harmonic fit is applied to remove the seasonality from the monthly data series. Then a multiple linear regression model is performed to solar activity, linear trend, and residual temperatures versus constant. Applying the regression analysis to latitude-averaged temperature and WV provides a more statistically significant value of their trends. Lyman-α flux is a proxy for solar**

activity, so the monthly mean of Lyman-α solar flux is used in multiple linear regression equation (1) as a measure of solar variability. Multiple linear regression analysis technique has been used by multiple authors in the past (e.g., Chandra et al., 1997; Hervig et al., 2015, 2016; Yue et al., 2019). To analyze the temperature and WV trends using multiple linear regression with the inclusion of the solar cycle and time we applied the following multiple regression analysis for trend estimation.

$$\text{Temperature or WV} = C_o + C_1(\text{Lyman}.\alpha) + C_2(\text{time}) + \text{error} \qquad (1)$$

Where $C_0$ is constant (intercept), $C_1$ and $C_2$ are regression coefficients characterizing the linear long-term trend (temperature and WV per year) and solar activity term. We calculate temperature and WV trends using multiple linear regression involving monthly temperature and WV (SABER) data over time. Before applying the multiple regression model we calculate solar radiation according to monthly data sets. For example, Monthly means of the Lyman-α index are computed for each month, yielding 176 points for both global and equator."

[Figure]

**A subpart of Figure 1:** Lyman- α index during January 2002-December 2023.

We have modified /revised the whole manuscript (including the language) based on the reviewers' comments.

A few examples of these updates are given below.

1. We revised all figures and tables.
2. We added additional required sections, for example, section 2.3 Solar cycle response, and section 2.3.1 Multiple linear regression analysis.
3. We removed less relevant (or having a repetition of information) sections, for example, section 3.1.3 (relationship between temperature and WV). We already showed an inverse relation between temperature and WV in multiple locations.
4. We rewrite whole sections, for example first section of the introduction part.
5. We revised the captions of all figures as suggested by the other reviewers.
6. We have made corrections in almost every line of the revised manuscript.

**Specific Comments**

**(R3-C2)** Page 1, lines 25-26: As discussed later, this statement does not take periodic forcing terms into account.

Response to (R3-C2): Agree, we have updated the sentences (after applying multiple linear regression) (lines 28-32 of the revised manuscript) and the same is given below

**"The temperature showed apparent positive responses to solar activity through all latitudes, which is more significant near the NH. Twenty-two years of monthly mean analysis shows that SH is warmer than NH during July and September and colder than NH during January and March. A cooling trend in temperature is in the range of 0.58 to 1.21 K/decade. Water vapor shows a positive trend and a strong negative correlation with the temperature."**

**(R3-C3)** Page 3, lines 83-86: These are old references that only address a specific location and time, and discuss altitudes below the mesopause region. Why are they cited?

Response to (R3-C3): Agree, the sentence removed from the revised manuscript (lines 80-82 of the revised manuscript) and the updated sentence is given below.

**"In the mesopause region, more abundant WV from increasing CH4 contributes to more frequent NLC occurrences (Lübken et al., 2018). WV content in the atmosphere controls the concentration of ozone that, in turn, affects mesospheric cooling (Smith, 2004)."**

Removed sentences are visible in the track change version of the manuscript (lines 123-126).

**(R3-C4)** Page 3, lines 94-95: But there are certainly long data sets from ground-based microwave measurements and satellites (e.g. HALOE, SOFIE) that should be discussed.

Response to (R3-C4): Agree, we have added relevant text and updated references related to ground-based microwave measurements and satellites as suggested (lines 85-100 of the revised manuscript) and the same is given below

**"Ground-based microwave radiometry is the ideal technique to monitor WV in the atmosphere, however, as compared to temperature; there have been fewer observations of WV in the upper mesosphere. There are certainly long data sets from ground-based microwave measurements and satellites. For example, long-term WV measurements made by the Sounding of the Atmosphere using the Broadband Emission Radiometry (SABER) instrument and by the Aura Microwave Limb Sounder instrument are analyzed by Yue et al. (2019). Nedoluha et al. (2022) presented ground-based microwave**

measurements of mesospheric WV made by the Water Vapor Millimeter-wave Spectrometer between 1992-2021. Straub et al. (2010) and Schranz et al. (2019) estimate the vertical gradient of WV inside of the polar vortex in autumn based on microwave radiometry measurements at polar latitudes. The Aura satellite with the Microwave Limb Sounder collects global WV profiles with coverage at a fixed local time due to its sun-synchronous orbit (Livesey et al., 2006). The dependence of NLCs on WV and temperature was quantified by Hervig et al. (2015), using Observations from the Solar Occultation For Ice Experiment (SOFIE). The latest progress and applications of atmospheric WV lidar calibration have been recently reviewed by Guo et al. (2024).".

If still we are missing any relevant references please let us know, will add them as suggested.

**References**

➢ Guo, X., Wu, D., Wang, Z., Wang, B., Li, C., Deng, Q., and Liu, D.: A review of atmospheric water vapor lidar calibration methods, Wiley Interdiscip. Rev. Water, 11, e1712, 2024.

➢ Hervig, M. E., Siskind, D. E., Bailey, S. M., and Russell III, J. M.: The influence of PMCs on water vapor and drivers behind PMC variability from SOFIE observations, J. Atmos. Solar-Terrestrial Phys., 132, 124–134, https://doi.org/https://doi.org/10.1016/j.jastp.2015.07.010, 2015.

➢ Livesey, N. J., Van Snyder, W., Read, W. G., and Wagner, P. A.: Retrieval algorithms for the EOS Microwave limb sounder (MLS), IEEE Trans. Geosci. Remote Sens., 44, 1144–1155, 2006.

➢ Nedoluha, G. E., Gomez, R. M., Boyd, I., Neal, H., Allen, D. R., Siskind, D. E., Lambert, A., and Livesey, N. J.: Measurements of mesospheric water vapor from 1992 to 2021 at three stations from the Network for the Detection of Atmospheric Composition Change, J. Geophys. Res. Atmos., 127, e2022JD037227, https://doi.org/https://doi.org/10.1029/2022JD037227, 2022.

➢ Schranz, F., Tschanz, B., Rüfenacht, R., Hocke, K., Palm, M., and Kämpfer, N.: Investigation of Arctic middle-atmospheric dynamics using 3 years of $H_2O$ and $O_3$ measurements from microwave radiometers at Ny-Ålesund, Atmos. Chem. Phys., 19, 9927–9947, 2019.

➢ Straub, C., Murk, A., and Kämpfer, N.: MIAWARA-C, a new ground based water vapor radiometer for measurement campaigns, Atmos. Meas. Tech., 3, 1271–1285, 2010

➢ Yue, J., Russell III, J., Gan, Q., Wang, T., Rong, P., Garcia, R., and Mlynczak, M.: Increasing water vapor in the stratosphere and mesosphere after 2002, Geophys. Res. Lett., 46, 13452–13460, https://doi.org/10.1029/2019GL084973, 2019.

**(R3-C5)** Page 4, lines 111-112: 40% is a substantial amount of missing data. Does this represent frequent small gaps, or less frequent large gaps? What dimension is used for averaging? Altitude? Time? What is the weighting function?

Response to (R3-C5): Sorry, ~40% was just an estimated number which was wrong. We have corrected the information in the revised manuscript (lines 146-148 of the revised manuscript) and the same is given below

**"We used 176 months of data for both the equator and the global mesopause during the study period. There are 44, and 46 missing months for NH and SH respectively (Figure 1a), and ~9% of missing months are filled with a weighted average."**

For a constant altitude (let's say 80 km), we have January data for all years 2002 to 2016, except January 2008 which is missing. In such case, we can fill January 2008 with the help of neighbor January values at the same altitude. We think this way of averaging is relatively more accurate than other techniques, and we applied this way of averaging on ~9% of missing data.

**(R3-C6)** Page 4, lines 120-122: This statement repeats the previous sentence. Since a 60-day yaw cycle is not an even fraction of a 365-day year, the latitude coverage in selected months will shift over 22 years. Are each of your months fully populated during the full data record?

Response to (R3-C6): Repeated information is removed from the revised manuscript. The section has been significantly improved and the required information is provided in detail (lines 126-175 of the revised manuscript) and the same is given below

**"2.2. TIMED-SABER instrument**

**The TIMED-SABER satellite views 90° to the right of the velocity vector of the TIMED spacecraft, and completes a full 24-hour local time coverage in 60-63 days (Russell III et al., 1999; Mlynczak et al., 2003; Figure 1). The SABER instrument scans the atmosphere from the troposphere up to the lower thermosphere and obtains vertical profiles kinetic temperature and volume mixing ratio of WV (Russell et al., 1999). The instrument performs near-global measurements and provides an excellent quality of the measured infrared limb radiances (Esplin et al., 2023). Technical description of the SABER instrument and further relevant information are discussed by Mlynczak, (1997) and Russell III et al. (1999)."**

Removed sentences are visible in the track change version of the manuscript (lines 199-202).

Details related to the yaw cycle and latitude coverage are given in the response to comment number 1 above. Let me put it again for you (Figure 1)

Revised Figure 1: (page # 7)

[Figure]

**Revised Figure 1**. SABER instrument latitude coverage versus time for observation. a) Monthly data coverage in selected months versus latitude ranges from January 2002 to December 2023, excluding transitional months. b) Comparison of SABER latitude coverage and monthly data versus time during years (2002-2003). c) Typical temporal coverage of TIMED-SABER instrument measurements. d) Latitude versus longitude tangent point locations for one day of observations in its north viewing phase (83°N to 52°S) – a north viewing yaw mode.

**(R3-C7)** Page 5, lines 133-135: Any comment on why the 2002 profile is such an outlier in the first 4 panels of Figure 2?

Response to (R3-C7):

The following may be the possible reasons.

First, the temperature is higher during the higher solar activity periods with the highest intensity observed during the period 2002–2003. This period represents the depletion phase of the 23$^{rd}$ solar cycle which was more active than the 24$^{th}$ solar cycle.

Second, during 2002 (July to September) high latitudes in the Southern Hemisphere experienced extreme planetary wave activity (unusual warming) throughout the vertical extent sampled by SABER (Kruger et al., 2005). Short-term variability in temperature is primarily due to small-scale gravity waves and tides (Dalin et al., 2017; Zhao et al., 2020). Gravity and planetary waves (Dalin et al., 2017), and atmospheric tides (Smith, 2004) bring periodic variations in temperature.

Third, some authors excluded January 2002 and they started sampling from February 2002 (Xu et al., 2007 and Zhao et al., 2020), and we have included January 2002, which may be the reason for the relatively high profile during 2002.

References

- Dalin, P., Kirkwood, S., Pertsev, N., and Perminov, V.: Influence of solar and lunar tides on the mesopause region as observed in polar mesosphere summer echoes characteristics, J. Geophys. Res. Atmos., 122, 10–369, https://doi.org/10.1002/2017JD026509, 2017.
- Krüger, K., Naujokat, B., & Labitzke, K. (2005). The unusual midwinter warming in the Southern Hemisphere stratosphere 2002: A comparison to Northern Hemisphere phenomena. Journal of the Atmospheric Sciences, 62(3), 603–613. https://doi.org/10.1175/JAS-3316.1
- Smith, A. K.: Physics and chemistry of the mesopause region, J. Atmos. solar-terrestrial Phys., 66, 839–857, https://doi.org/10.1016/j.jastp.2004.01.032, 2004.
- Zhao, X. R., Sheng, Z., Shi, H. Q., Weng, L. B., and Liao, Q. X.: Long-term trends and solar responses of the mesopause temperatures observed by SABER during the 2002–2019 period, J. Geophys. Res. Atmos., 125, e2020JD032418, https://doi.org/10.1029/2020JD032418, 2020.

**(R3-C8)** Pages 5-6, lines 138-143: There is a significant and well-known solar cycle signal in mesospheric temperature (noted on lines 242-249) that will greatly affect any calculated trends. Your results (which do not include any uncertainty estimate) cannot be compared to previous trends unless this contribution is addressed.

Response to (R3-C8): Agree, we have included solar cycle effects and revised all results (Pages 117 and onward of the revised manuscript) and the same is given below

**"2.3. Solar cycle response**

**A regular variation in the occurrence of the Sun's active regions, with a periodicity of ~11- years, is called the solar cycle. Radio waves emissions from the Sun vary with the solar cycle and are enhanced (radio burst) during chromospheric or coronal events.**

Since these emissions can easily be recorded (e.g., at 10.7 cm or 2.8 GHz), they are often used as an indicator of solar activity. Over the 11-year solar cycle, the solar flux varies by a factor of 2 at Lyman-α (Brasseur and Solomon, 2005). The 11-year solar cycle (Lean et al., 1997), has a direct impact on the upper atmosphere ( mesopause region). At the mesospheric height, WV is strongly photo-dissociated by solar Lyman-α (Brasseur and Solomon, 1986). Ultraviolet radiation from the Sun is enhanced during the maximum of the 11-year solar cycle. Solar cycle (temperature) variations impact on WV and their relationship has been quantified by multiple researchers in the past (Brasseur and Solomon, 1986; Chandra et al., 1997; Fiedler et al., 2011; Hervig and Siskind, 2006; Siskind et al., 2013). In the present work, we include the Lyman-α solar index, the local time, seasonal and solar cycle variations of temperature, and WV during January 2002-December 2023. The Lyman-α solar index is obtained from the OMNIWeb database (https://omniweb.gsfc.nasa.gov). Figure 1(e) shows Lyman-alpha variation every month during the study period. The 23rd solar cycle ended in December 2008 and 24th solar cycle began (Figure 1e), and was minimal up to 2010. Consequently, the Lyman-□ was observed to be lower during solar minima. The period following 2010 is the solar active period of the 24[th] solar cycle and the corresponding variation in Lyman-α can be noticed in Figure 1e. "

Trends significantly reduced after including solar cycle impact (using multiple linear regression analysis) as mentioned against the response of comment # 1 and the same is given below.

"2.3.1. Multiple linear regression analysis

To investigate the long-term trends (temperature and WV) and the solar response of the mesopause temperature, a three-component harmonic fit is applied to remove the seasonality from the monthly data series. Then a multiple linear regression model is performed to solar activity, linear trend, and residual temperatures versus constant. Applying the regression analysis to latitude-averaged temperature and WV provides a more statistically significant value of their trends. Lyman-α flux is a proxy for solar activity, so the monthly mean of Lyman-α solar flux is used in multiple linear regression equation (1) as a measure of solar variability. Multiple linear regression analysis technique has been used by multiple authors in the past (e.g., Chandra et al., 1997; Hervig et al., 2015, 2016; Yue et al., 2019). To analyze the temperature and WV trends using multiple linear regression with the inclusion of the solar cycle and time we applied the following multiple regression analysis for trend estimation.

$$\text{Temperature or WV} = C_o + C_1(\text{Lyman}.\,\alpha)\ \ + C_2(\text{time}) + \text{error} \tag{1}$$

Where $C_0$ is constant (intercept), $C_1$ and $C_2$ are regression coefficients characterizing the linear long-term trend (temperature and WV per year) and solar activity term. We calculate temperature and WV trends using multiple linear regression involving monthly temperature and WV (SABER) data over time. Before applying the multiple regression model we calculate solar radiation according to monthly data sets. For example, Monthly means of the Lyman-α index are computed for each month, yielding 176 points for both global and equator."

[Figure]

**A subpart of Figure 1:** Lyman- α index during January 2002-December 2023.

**(R3-C9)** Page 6, lines 154-156: This statement seems very simplistic, given that your "global" average only includes small latitude bands near each pole and at the Equator. There is also significant altitude dependence that can vary between months and years.

Response to (R3-C9): In the revised manuscript we have removed the mentioned statement (visible in the track change version). The global mesopause term used in this article does not include small latitude bands near each pole and at the equator but includes all latitude ranges of the mesopause scanned by the SABER instrument. This is one of the maximum spatial coverage covering both the hemispheres alternatively. Still, there are some limitations and uncertainties on the global scale that are related to the yaw cycle/mixing of northern and southern hemisphere data, etc. We have mentioned these limitations in the last section of the revised manuscript (lines 126-175 of the revised manuscript) and the same is given below

**"2.2. TIMED-SABER instrument**

**The TIMED-SABER satellite views 90° to the right of the velocity vector of the TIMED spacecraft, and completes a full 24-hour local time coverage in 60-63 days (Russell III et al., 1999; Mlynczak et al., 2003; Figure 1). The SABER instrument scans the atmosphere from the troposphere up to the lower thermosphere and obtains vertical profiles kinetic temperature and volume mixing ratio of WV (Russell et al., 1999). The instrument performs near-global measurements and provides an excellent quality of the measured infrared limb radiances (Esplin et al., 2023). Technical description of the SABER instrument and further relevant information are discussed by Mlynczak, (1997) and Russell III et al. (1999). TIMED satellite rotates 180° about its yaw axis and provides latitude coverage continuously in the range of 53°S to 83°N and then switching to 83°S to 53°N every ~60 days (Russell III et al., 1999). Due to the asymmetrical latitudinal coverage of the SABER instrument, there are some missing measurement months at high latitudes (52°N-83°N or 52°S-83°S). Multiple studies ( e.g; Forbes et al., 2021; Liu et al., 2017; Das, 2021) are limited to the latitude band ~50°S to ~50°N, mainly due to the TIMED ~60 days yaw cycle. In the present study, we have included high-latitude regions from both hemispheres along with some missing data. For example, coverage of high northern latitudes included July in the early years, but not during the recent several years (2017-2023)………"** Please have a look at this section in the revised manuscript for full details.

Revised Figure 1: (==page # 7==)

[Figure]

**Revised Figure 1**. SABER instrument latitude coverage versus time for observation. a) Monthly data coverage in selected months versus latitude ranges from January 2002 to December 2023, excluding transitional months. b) Comparison of SABER latitude coverage and monthly data versus time during years (2002-2003). c) Typical temporal coverage of TIMED-SABER instrument measurements. d) Latitude versus longitude tangent point locations for one day of observations in its north viewing phase (83°N to 52°S) – a north viewing yaw mode.

Uncertainties related to high latitude regions: (line numbers 713 and onward )

**"Section 6. Associated uncertainties and limitations**

**The possible sources of uncertainties during the analysis of long-term temperature and WV are mentioned below.**

1. **Large uncertainty is related to the analysis of temperature and WV over SH and NH (above ~53⁰ latitudes) and has a relatively larger bias in results as compared to the results over the equator. The yaw cycle is ~60 days, and only one polar region (SH or NH) is observed in each yaw cycle, and the selected polar regions are only alternatively observed half of a year owing to the yawing of the TIMED satellite. In other words, the latitudinal coverage is governed by a 60-day yaw cycle that allows observations of latitudes from 83⁰S to 52⁰N in the south-viewing phase or from 53⁰S to 82⁰N in the North-viewing phase (further details are given in the text). Multiple studies (e.g; Forbes et al., 2021; Liu et al., 2017; Mlynczak et al., 2022; Das, 2021) are limited to the latitude band ~50°S to ~50°N. In the present study, we have included high-latitude regions from both hemispheres along with some missing months. Missing months are usually April, August, or December in the NH and February, June, or October in the SH. As a result, the choice of these months for high latitudes introduces a systematic bias in the time series.**

2. **Temperature and WV trends over NH and SH are calculated for six months because April and December data were insufficient for long-term trends over NH. Similarly, June and October data was limited for SH trend estimation. Therefore, trends over the equator are more accurate than those of NH and SH trends.**
   **"**

So, we present our results along with the above-mentioned uncertainties in the revised text.

**(R3-C10)** Page 8, lines 201-204: The large variation in water vapor mixing ratio over this altitude range (as shown in Figures 2b and 2f) means that a simple average will be dominated by values from the lowest portion of the profile.

Response to (R3-C10): Agree this is a source of uncertainty for the current article and we have mentioned it in the revised manuscript (lines 747-749 of the revised manuscript) and the same is given below.

**"The large variation in the WV mixing ratio over the mesopause altitude range (as shown in Figure 2) means that our simple average can be dominated by values from the lowest portion of the profile."**

We have introduced a new section (Associated uncertainties and limitations) in the revised manuscript and and above text related to simple averaging is one of its points.

**(R3-C11)** Pages 8-9, lines 204-206: Solar activity-induced variations will greatly affect any calculated trends in water vapor, as discussed previously for your temperature analysis.

Response to (R3-C11): Agree, we have included solar cycle effects (lines 176-196 of the revised manuscript) and revised all results and the same is given below

**"2.3. Solar cycle response**

**A regular variation in the occurrence of the Sun's active regions, with a periodicity of ~11- years, is called the solar cycle. Radio waves emissions from the Sun vary with the solar cycle and are enhanced (radio burst) during chromospheric or coronal events. Since these emissions can easily be recorded (e.g., at 10.7 cm or 2.8 GHz), they are often used as an indicator of solar activity. Over the 11-year solar cycle, the solar flux varies by a factor of 2 at Lyman-α (Brasseur and Solomon, 2005). The 11-year solar cycle (Lean et al., 1997), has a direct impact on the upper atmosphere ( mesopause region). At the mesospheric height, WV is strongly photo-dissociated by solar Lyman-α (Brasseur and Solomon, 1986). Ultraviolet radiation from the Sun is enhanced during the maximum of the 11-year solar cycle. Solar cycle (temperature) variations impact on WV and their relationship has been quantified by multiple researchers in the past (Brasseur and Solomon, 1986; Chandra et al., 1997; Fiedler et al., 2011; Hervig and Siskind, 2006; Siskind et al., 2013). In the present work, we include the Lyman-α solar index, the local time, seasonal and solar cycle variations of temperature, and WV during January 2002- December 2023. The Lyman-α solar index is obtained from the OMNIWeb database (https://omniweb.gsfc.nasa.gov). Figure 1(e) shows Lyman-alpha variation every month during the study period. The 23$^{rd}$ solar cycle ended in December 2008 and 24$^{th}$ solar cycle began (Figure 1e), and was minimal up to 2010. Consequently, the Lyman-α was observed to be lower during solar minima. The period following 2010 is the solar active period of the 24$^{th}$ solar cycle and the corresponding variation in Lyman-α can be noticed in Figure 1e. "**

Trends significantly reduced after including solar cycle impact (using multiple linear regression analysis) as mentioned against the response of comment # 1 and the same is given below.

**"2.3.1. Multiple linear regression analysis**

To investigate the long-term trends (temperature and WV) and the solar response of the mesopause temperature, a three-component harmonic fit is applied to remove the seasonality from the monthly data series. Then a multiple linear regression model is performed to solar activity, linear trend, and residual temperatures versus constant. Applying the regression analysis to latitude-averaged temperature and WV provides a more statistically significant value of their trends. Lyman-α flux is a proxy for solar activity, so the monthly mean of Lyman-α solar flux is used in multiple linear regression equation (1) as a measure of solar variability. Multiple linear regression analysis technique has been used by multiple authors in the past (e.g., Chandra et al., 1997; Hervig et al., 2015, 2016; Yue et al., 2019). To analyze the temperature and WV trends using multiple linear regression with the inclusion of the solar cycle and time we applied the following multiple regression analysis for trend estimation.

$$\text{Temperature or WV} = C_o + C_1(Lyman.\alpha) \; + \; C_2(time) + error \qquad (1)$$

Where $C_0$ is constant (intercept), $C_1$ and $C_2$ are regression coefficients characterizing the linear long-term trend (temperature and WV per year) and solar activity term. We calculate temperature and WV trends using multiple linear regression involving monthly temperature and WV (SABER) data over time. Before applying the multiple regression model we calculate solar radiation according to monthly data sets. For example, Monthly means of the Lyman-α index are computed for each month, yielding 176 points for both global and equator."

[Figure]

**A subpart of Figure 1:** Lyman- α index during January 2002-December 2023.

**(R3-C12)** Page 9, lines 224-225: Where is this result shown?

Response to (R3-C12): We provide a relevant source of the information (lines 319-320 of the revised manuscript) and the same is given below

**"WV in the polar region is relatively higher in summer than in winter (Hervig et al., 2003). This may be due to upwelling transport in the summer hemisphere from lower altitudes towards the mesopause (Körner and Sonnemann, 2001). "**

References

Hervig, M., McHugh, M., and Summers, M. E.: Water vapor enhancement in the polar summer mesosphere and its relationship to polar mesospheric clouds, Geophys. Res. Lett., 30, https://doi.org/10.1029/2003GL018089, 2003.

Körner, U. and Sonnemann, G. R.: Global three-dimensional modeling of the water vapor concentration of the mesosphere-mesopause region and implications with respect to the noctilucent cloud region, J. Geophys. Res. Atmos., 106, 9639–9651, https://doi.org/10.1029/2000JD900744, 2001

**(R3-C13)** Pages 9-10, lines 237-238: But Figure 2 only shows global averages, not individual latitude ranges.

Response to (R3-C13): We have revised Figure 2 and have included additional latitude ranges. The mentioned sentence does not fit with the flow of information and is deleted from the revised text (visible in the track change version). The  revised Figure 2 (Page number 11) is given below

[Figure]

Figure 2. Temperature and water vapor gradient between 80-100 km altitudes from SABER observations at the three selected latitude bins during 200-2023. a) Equator (0° ± 1º). b) Northern hemisphere (80°N ± 1º). (c) Southern hemisphere (80°S ± 1º), in the indicated months, by averaging all January, June, and September values from 2002 to 2023.

**(R3-C14)** Page 10, lines 238-239: Again, Figure 2 only shows results averaged over 80-100 km, so what is the basis for this statement?

Response to (R3-C14): We have revised Figure 2 by including additional months from selected three latitude bins (page 11 of the revised manuscript) as shown in the above response.

The basis of the mentioned sentence was below Figure which is not shown in the manuscript.

[Figure]

**(R3-C15)** Page 10, lines 252-254: There are numerous studies during the last 30 years with more advanced models.

Response to (R3-C15): Agree, we have used/cited relatively latest references (eg: Berger and von Zahn, 2002; Korner and Sonnemann, 2001; von Zahn and Berger, 2003; Grygalashvyly and Sonnemann, 2006; Rong et al., 2012) in multiple places of the revised text and the same is given below.

Lines (333 of the revised manuscript)
**"Models suggest that the water vapor mixing ratio at 90 km altitude is around 1–2 ppmv (Berger and von Zahn, 2002; Korner and Sonnemann, 2001; von Zahn and Berger, 2003)."**

Lines (352 of the revised manuscript)
**"Korner and Sonnemann, (2001) demonstrated that values of the mean zonal component of vertical wind velocity reach about 1-2 cm s−1 at 80-90 km altitude."**

References

Berger, U. and Von Zahn, U.: Icy particles in the summer mesopause region: Three-dimensional modeling of their environment and two-dimensional modeling of their transport, J. Geophys. Res. Sp. Phys., 107, SIA-10, https://doi.org/10.1029/2001JA000316, 2002.

Grygalashvyly, M. and Sonnemann, G. R.: Trends of mesospheric water vapor due to the increase of methane–A model study particularly considering high latitudes, Adv. Sp. Res., 38, 2394–2401, https://doi.org/10.1006/j.asr.2006.09.010, 2006.

Körner, U. and Sonnemann, G. R.: Global three-dimensional modeling of the water vapor concentration of the mesosphere-mesopause region and implications with respect to the noctilucent cloud region, J. Geophys. Res. Atmos., 106, 9639–9651, https://doi.org/10.1029/2000JD900744, 2001.

Rong, P. P., Russell III, J. M., Hervig, M. E., and Bailey, S. M.: The roles of temperature and water vapor at different stages of the polar mesospheric cloud season, J. Geophys. Res. Atmos., 117, https://doi.org/10.1029/2011JD016464, 2012, 2012.

Von Zahn, U. and Berger, U.: Persistent ice cloud in the midsummer upper mesosphere at high latitudes: Three-dimensional modeling and cloud interactions with ambient water vapor, J. Geophys. Res. Atmos., 108, https://doi.org/10.1029/2002JD002409, 2003.

 **(R3-C16)** Page 10, line 257: Table 1 mixes different selections for latitude coverage, seasonal coverage, long-term temporal coverage, and data sources. It is hard to know what conclusions could (or should) be drawn.

Response to (R3-C16): We have revised the Table as suggested by the other two reviewers (pages number 14, 15, and 16 of the revised manuscript) and the same is given below. In comparison to the previous table, the revised version of Table 1 has detailed and relevant information, please have a look.

Table 1. Temperature and water vapor content comparisons with past studies in mesopause

| Trend K/decade | Avg. Temp | Altitude (km) | Location/Season/Data source | References |
|---|---|---|---|---|
| Temperature | | | | |
| Min: | 184.54 K | 80-100 | Global/summer (Jun. and Jul.)/ SABER | This study |
| 0 | 188.20 K | | Global/winter (Jan. and Dec.)/ SABER | |
| | 162.64 K | | 80°N ± 1º / summer (Jun. and Jul.)/SABER | |
| | 201.14 K | | 80°N ± 1º / winter (Jan. and Dec.)/SABER | |
| Max: | 193.21 K | | 80°S ± 1º / summer (Jun. and Jul.)/SABER | |
| -1.21 | 161.14 K | | 80°S ± 1º / winter (Jan. and Dec.)/SABER | |
| | 185.81 K | | 0° ± 1º / summer (Jun. and Jul.)/SABER | |
| | 188.95 K | | 0° ± 1º / winter (Jan. and Dec.)/SABER | |
| Min: 0 | 130-190K | 80-100 | 83°N to 83°S- all latitudes/ SABER | Zhao et al., 2020 |
| | 188±2 K | | 83°N/Northern hemisphere / SABER | |
| Max: | 135±2 K | | 83°S/Southern hemisphere / SABER | |
| -1.4 | 158±2 K | | 0°/Equator / SABER | |
| | 139 K | 90 | 80°S/January/ SABER | Wang et al., 2022 |
| | 180 K | 86 | 40°S/January/ SABER | |
| | 129 K | 90 | 80°N/July/ SABER | |
| | 161 K | 83 | 55°N/July/ SABER | |
| | 160 K | ~100 | 30°N/ around equinoxes (March)/ SABER | Xu et al., 2007 |
| | 185 K | ~80 | 30°N/ around equinoxes (March)/ SABER | |
| | 124 K | ~100 | 80°N /solstice period (June)/ SABER | |
| | 135 K | ~80 | 80°N /solstice period (June)/ SABER | |
| | 133 K | ~100 | 80°N /solstice period (December)/ SABER | |
| | 143 K | ~80 | 80°N /solstice period (December)/ SABER | |

| | | | | |
|---|---|---|---|---|
| | ~126 K | 80-100 | Summer polar region/ SABER | |
| | ~190 K | 80-100 | Winter polar region/ SABER | |
| | 156-162 K | 84 | 45–50°N/ summer night time (Aura/MLS) | Dalin et al., 2023 |
| | 152-157 K | 84 | 50–55°N/ summer night time (Aura/MLS) | |
| | 147-151 K | 84 | 55–60°N/ summer night time (Aura/MLS) | |
| | 151-159 K | 89 | 45–50°N/ summer night time (Aura/MLS) | |
| | 147-153 K | 89 | 50–55°N/ summer night time (Aura/MLS) | |
| | 141-146 K | 89 | 55–60°N/ summer night time (Aura/MLS) | |
| -2.5 | ~177.6 K | 97 | 41°N - 42°N / non summer months /Na lidar | Yuan 2019 |
| -2.3 | ~177.6 K | 92 | 41°N- 42°N /non winter months / Na lidar | |
| -3.8 | ~177.6 K | 97 | 41°N - 42°N / winter /Na lidar | |
| -1.75 | ~177.6 K | 92 | 41°N- 42°N /summer / Na lidar | |
| -2.3 | 160-230 K | 87 | 51°N/ all seasons/SABER instrument | Offermann et al., 2010 |
| Up | 158-238 K | 87 | 51°N/ all seasons/OH | |
| To | 160-232 K | 87 | 48°N/ all seasons/SABER instrument | |
| -6.0 | 145-235 K | 87 | 48°N/ all seasons/ OH | |
| -6.8 | | ~100 | 41°N (Lidar + SABER + Model) | She et al., 2009 |
| -1.5 | ~184K | ~91 | 41°N/January  (Lidar + SABER + Model) | |
| - 0.64 | ~200 K | ~85 | 41°N/January  Na lidar | |
| -0.64 | 160-245 K | 85-86 | 41°N/all seaons/Na lidar | She et al., 2015 |
| -2.8 | 160-235 K | 91-93 | 42°N/all seasons/Na lidar | |
| -0.23 | | 87 | 69°S/winter /Hydroxyl airglow | French et al., 2005 |
| -0.5 | | 80-95 | ±52° latitude  (WACCM-Model) | Garcia et al., 2019 |
| -2.4 | 160-173 K | 80-100 | ~57°N / summer mesopause (ground-based) | Dalin et al., 2020 |
| -0.4 | 202-218 K | 80-100 | ~57°N/winter (ground-based) | |
| -0.89 | 194-202 K | 87 | 51° N/annual mean | Kalicinsky et al., 2016 |
| | 185-201 K | 87 | 48∘ N/annual mean | |
| -4.0 | 135 K | 90 | 78ⁿN/summer MLS on the Aura satellite. | Hall et al., 2012 |
| | ~200 K | 90 | 78ⁿN/winter / radar observation | |
| -1.2 | 146-154 K | 83 | 55–61°N/ annual /LIMA and MIMAS model | Lübken et al., 2018 |
| -2.9 | 160-230 K | 98.5 | 41°N/all season/ Na lidar | She and Krueger, 2004 |
| -2.1 | 198-228 K | 80-100 | 63°N/ January/ SABER | Ammosov et al., 2014 |
| | 196-215 K | 80-100 | 63°N/ February/ SABER | |
| -2 | | 80-100 | middle & subpolar latitudes /summer/ model | Grygalashvyly et al., 2014 |
| -0.5 | | | middle & subpolar latitudes/ winter/ model | |
| -2.2 | | 80-100 | Middle latitudes/Airglow measurement | Perminov et al., 2014 |
| -0.24 | 140-170 K | 80 - 84 | 64–74°N/ all season/SOFIE | Hervig et al., 2015 |
| -0.5 | 145-166 K | 80-84 | 77°N /Satellite instrument and Model | Hervig et al., 2016 |
| -1.2 | 140-220 K | 87 | 68°S/ winter/ OH nightglow | French et al., 2020a |
| -0.3 | ~196 K | 87 | 23-26°S /March-April/SABER & airglow | Noll et al., 2017 |
| | 145-235 K | ~87 | 74°N /spectrometric observations of the OH | Medvedeva and Ratovsky, 2023 |

**Water vapor mixing ratio**

| | | | | |
|---|---|---|---|---|
| ~1.30 ppmv | 80-100 | Global/summer/ SABER | This study | |
| ~1.20 ppmv | | Global/winter/ SABER | | |
| ~1.90 ppmv | | 80°N ± 1º /summer/SABER | | |
| ~0.49 ppmv | | 80°N ± 1º /winter/SABER | | |
| ~0.67 ppmv | | 80°S ± 1º / summer/SABER | | |
| ~2.30 ppmv | | 80°S ± 1º /winter/SABER | | |
| ~1.20 ppmv | | 0° ± 1º /summer/SABER | | |
| ~1.10 ppmv | | 0° ± 1º /winter/SABER | | |
| 4.2-5.1 ppmv | 84 | 45–50°N/ summer night time (Aura/MLS) | Dalin et al., 2023 | |
| 4.5-5.4 ppmv | 84 | 50–55°N/ summer night time (Aura/MLS) | | |
| 4.7-5.6 ppmv | 84 | 55–60°N/ summer night time (Aura/MLS) | | |
| 3.1-3.6 ppmv | 89 | 45–50°N/ summer night time (Aura/MLS) | | |
| 3.3-3.9 ppmv | 89 | 50–55°N/ summer night time (Aura/MLS) | | |
| 3.3-3.9 ppmv | 89 | 55–60°N/ summer night time (Aura/MLS) | | |
| 1-8 ppmv | 80 - 84 | 64–74°N/ all season/SOFIE | Hervig et al., 2015 | |
| 5.4-5.8 ppmv | 80-84 | 77°N /Satellite instrument and Model | Hervig et al., 2016 | |
| 0-7.0 ppmv | 80-100 | 66°-79°N/SOFIE on AIM & ALOMAR lidar | Hervig et al., 2009a | |
| 1-2 ppmv | 95 | 66°-79°N/satellite measurement | | |

| | | | |
|---|---|---|---|
| 1 ppmv | 90 | 78°N/ summer/1-D model | Murray and Jensen, 2010 |
| 3 ppm | 86 | 67.9°N- Polar region/Summer/Model | Gumbel et al., 2003 |
| 2.3 ppmv | 85 | Mid-latitude /Jul./Ground-based microwave | Bevilacqua et al., 1983 |
| 1.6 ppmv | 85 | Mid-latitude /Sep./Ground-based microwave | |
| 1.0 ppmv | 85 | Mid-latitude /Jan./Ground-based microwave | |
| 1.2 ppmv | 85 | Mid-latitude /Apr./Ground-based microwave | |
| 0.1 ppmv | 85 | Mid-latitude /Dec./Ground-based microwave | |
| 1.1 ppmv | 85 | Mid-latitude /Apr./Ground-based microwave | |
| 0-7.0 ppmv | 80-100 | 66°-79°N/SOFIE on AIM & ALOMAR lidar | Hervig et al., 2009a |
| 1-2 ppmv | 95 | 66°-79°N/satellite measurement | |
| 1.5-4.5 ppmv | 80 | 69ºN/Ground-based microwave | Seele and Hartogh, 1999 |
| 2-2.5 ppmv | 85 | Polar summer/Jun., Jul., Aug.,/ground-based | |
| 0.2 ppmv | 84 | 67ºN/3-D model / | Von Zahn & Berger, 2003 |
| 3 ppmv | 80-83 | 50ºN-80ºN/3-D model / | |
| ~2.0 ppmv | 80-83 | 50ºN-80ºN/3-D model / | |
| ~1.5 ppm | 90 | 72.5ºN /Jul., Aug./3D-Model and HALOE | Körner & Sonnemann, 2001 |
| ~3.5 ppm | 85 | 72.5ºN /Jul., Aug./3D-Model and HALOE | |
| ~5.1 ppm | 80 | 72.5ºN /Jul., Aug./3D-Model and HALOE | |
| 1.0 ppmv | ~83 | 65°–70°N /winter / HALOE measurement | Hervig et al., 2003 |
| 8.0 ppmv | ~83 | 65°–70°N /summer / HALOE measurement | |
| 0.45 - 4.81 ppmv | 80-94 | 78°N/ summer/Model, | Lubken et al., 2004 |
| ~4.5 ppmv | 80 | 78°N/ summer/Model, | |
| 3.4 ppmv | 85 | 78°N/ summer/Model, | |
| 1.98 ppmv | 90 | 78°N/ summer/Model, | |
| 2-4 ppmv | 82 | 55°N–55°S/SABER | Yue et al., 2019 |
| ~3.5 ppmv | 80 | 19.5°N/ Sep./Spectrometer mouna | Nedoluha et al., 2022 |

**(R3-C17)** Page 12, lines 280-283: This statement says that you have confirmed previous work. Any new information?

Response to (R3-C17): Agree, the mentioned statement is relevant for another section 2.3 "Solar cycle response", and moved the statement from the previously mentioned location to section 2.3. We removed the whole section because the section had little new information.

**(R3-C18)** Page 15, lines 335-338: See previous questions about shift in yaw dates during SABER mission and impact on sampling. Note that the April data in Figure 5b only begin in 2017. Is the large trend in September temperature and water vapor affected by sampling changes?

Response to (R3-C18): Shift in yaw dates, SABER instruments latitudes and time relation, and Monthly data coverage are discussed in detail in the revised manuscript (lines 126 and onward of the revised manuscript).

**"2.2. TIMED-SABER instrument**

**The TIMED-SABER satellite views 90° to the right of the velocity vector of the TIMED spacecraft, and completes a full 24-hour local time coverage in 60-63 days (Russell III et al., 1999; Mlynczak et al., 2003; Figure 1). The SABER instrument scans the atmosphere from the troposphere up to the lower thermosphere and obtains vertical profiles kinetic temperature and volume mixing ratio of WV (Russell et al., 1999). The instrument performs near-global measurements and provides an excellent quality of the**

measured infrared limb radiances (Esplin et al., 2023). Technical description of the SABER instrument and further relevant information are discussed by Mlynczak, (1997) and Russell III et al. (1999). TIMED satellite rotates 180° about its yaw axis and provides latitude coverage continuously in the range of 53°S to 83°N and then switching to 83°S to 53°N every ~60 days (Russell III et al., 1999). Due to the asymmetrical latitudinal coverage of the SABER instrument, there are some missing measurement months at high latitudes (52°N-83°N or 52°S-83°S). Multiple studies ( e.g; Forbes et al., 2021; Liu et al., 2017; Das, 2021) are limited to the latitude band ~50°S to ~50°N, mainly due to the TIMED ~60 days yaw cycle. In the present study, we have included high-latitude regions from both hemispheres along with some missing data. For example, coverage of high northern latitudes included July in the early years, but not during the recent several years (2017-2023)………" Please have a look at this section in the revised manuscript for full details.

Revised Figure 1: (page # 7 )

[Figure]

**Revised Figure 1**. SABER instrument latitude coverage versus time for observation. a) Monthly data coverage in selected months versus latitude ranges from January 2002 to December 2023, excluding transitional months. b) Comparison of SABER latitude coverage and monthly data versus time during years (2002-2003). c) Typical temporal coverage of TIMED-SABER instrument measurements. d) Latitude versus longitude tangent point locations for one day of observations in its north viewing phase (83°N to 52°S) – a north viewing yaw mode.

We are sorry to say that previous trends were not correct and new trends are used in the text. Revised trends are given below

[Figure]

Figure 9. Temperature and water vapor trends during selected four months as indicated. a) Temperature and WV trends over NH (80⁰N ± 1⁰). b) Temperature and WV trends over SH (80⁰S ± 1⁰). c) Temperature and WV trends over the equator (0⁰ ± 1⁰). d) Temperature and WV trends on the global scale.

**(R3-C19)** Page 17, lines 350-354: Water vapor content at these low altitudes (81-83 km) will be affected by sublimation of PMC particles that settle from higher altitudes.

Response to (R3-C19): Thank you, we have provided additional text as suggested (lines 469-470 of the revised manuscript) and the same is given below

**"WV content at low altitudes (81-83 km) is affected by the sublimation of polar mesospheric cloud particles that settle from higher altitudes. The impact of polar mesospheric clouds on the surrounding WV (via dehydration and sublimation) is discussed by Hervig et al. (2015) and by Lübken and Berger, (2011)."**

References

Hervig, M. E., Siskind, D. E., Bailey, S. M., and Russell III, J. M.: The influence of PMCs on water vapor and drivers behind PMC variability from SOFIE observations, J. Atmos. Solar-Terrestrial Phys., 132, 124–134, https://doi.org/10.1016/j.jastp.2015.07.010, 2015

Lübken, F. and Berger, U.: Latitudinal and interhemispheric variation of stratospheric effects on mesospheric ice layer trends, J. Geophys. Res. Atmos., 116, 2011.

**(R3-C20)** Page 17, lines 375-376: Note that January/December is summer in the Southern Hemisphere, not winter.

Response to (R3-C20): Sorry for this mistake, we have updated the sentences as suggested (lines 492-494 of the revised manuscript) and the same is given below

**"During summer (January/December), temperature first decreased up to 92 km altitude and then increased to 93 km and onward altitudes."**

**(R3-C21)** Page 27, line 586: It's not clear why you say "performed the measurements" when the paper only analyzes SABER data.

Response to (R3-C21): Agree, we have modified the sentences as suggested (line 807 of the revised manuscript) and the same is given below

**"CG and SK initiated the idea; CG and DG analyzed and required calculations; CG, DG, and YY wrote the manuscript draft; SK and XG reviewed and edited the manuscript."**

Thanks to anonymous reviewer 3 for his/her constructive comments and suggestions.
----------------------- End of the response to reviewer 3 -----------------------